**Subject Category:**
Biology (whole organism)

ecology/environmental science/evolution

carnivores, density-dependence, density-dependent habitat selection, habitat selection functional response, recolonization, resource selection function

**Author for correspondence:**
Shawn T. O'Neil
e-mail: stoneil@mtu.edu;
oneil.shawnt@gmail.com

# Territorial landscapes: incorporating density-dependence into wolf habitat selection studies

Shawn T. O'Neil[1], Dean E. Beyer Jr[1,2]
and Joseph K. Bump[3]

[1]School of Forest Resources and Environmental Science, Michigan Technological University, 1400 Townsend Avenue, Houghton, MI, USA
[2]Wildlife Division, Michigan Department of Natural Resources, Marquette, MI, USA
[3]Department of Fisheries, Wildlife and Conservation Biology, University of Minnesota, St Paul, MN, USA

 STO, 0000-0002-0899-5220; JKB, 0000-0002-4369-7990

Habitat selection is a process that spans space, time and individual life histories. Ecological analyses of animal distributions and preferences are most accurate when they account for inherent dynamics of the habitat selection process. Strong territoriality can constrain perception of habitat availability by individual animals or groups attempting to colonize or establish new territory. Because habitat selection is a function of habitat availability, broad-scale changes in habitat availability or occupancy can drive density-dependent habitat functional responses. We investigated density-dependent habitat selection over a 19-year period of grey wolf (*Canis lupus*) recovery in Michigan, USA, using a generalized linear mixed model framework to develop a resource selection probability function (RSPF) with habitat coefficients conditioned on random effects for wolf packs and random year intercepts. In addition, we allowed habitat coefficients to vary as interactions with increasing wolf density over space and time. Results indicated that pack presence was driven by factors representing topography, human development, winter prey availability, forest structure, roads, streams and snow. Importantly, responses to many of these predictors were density-dependent. Spatio-temporal dynamics and population changes can cause considerable variation in wildlife–habitat relationships, possibly confounding interpretation of conventional habitat selection models. By incorporating territoriality into an RSPF analysis, we determined that wolves' habitat use in Michigan shifted over time, for example, exhibiting declining responses to winter prey indices and switching from positive to negative responses with respect to stream densities. We consider

this an important example of a habitat functional response in wolves, driven by colonization, density-dependence and changes in occupancy during a time period of range expansion and population increase.

## 1. Introduction

Investigating an organism's habitat preference and quantifying its realized niche is fundamental for ecologists (*sensu* [1–4]). Population ecology and conservation biology rely on habitat selection studies because identifying the factors influencing distributions, densities, gene flow and fitness characteristics in species is necessary to manage populations and conserve habitat. Resource or habitat selection functions (RSFs or HSFs; [5]), species distribution models and step selection analysis [6,7] are all broadly used to explore biotic elements that drive habitat use and species range [8]. These modelling approaches have contributed greatly to our understanding of animal–habitat relationships, but applications are often limited in terms of predictive and explanatory capacity for a variety of reasons [9,10].

Habitat selection is a behavioural process that spans multiple dimensions [11–13]. Ecological analyses of animal distributions and preferences are most accurate when they incorporate the inherent dynamics of the habitat selection process [14,15]. Habitat selection is a function of habitat availability (a habitat functional response; [16,17]), and changes in availability may coincide with variation in the local population density of inter- or intraspecific individuals [15,18]. Changes in habitat availability can also occur due to environmental stochasticity (e.g. drought) or human impacts (e.g. land cover change). The degree to which functional responses are linked to density probably depends on the nature of competitive behaviour between or among species. Evidence suggests that ideal-free consumers generalize their habitat selection with increases in conspecific density [19–21]. In this case, the distribution of available habitat may change in composition but geographical availability can remain constant. However, territorial species are not as well understood and the implication of increasing density on corresponding availability distributions has not been addressed in detail from a habitat modelling perspective.

According to density-dependent habitat selection theory, both territorial and non-territorial species should increase their use of suboptimal habitat patches as density increases [22–24]. In populations that select habitat according to an ideal-free distribution (i.e. non-territorial behaviour), variation in density theoretically reflects the quality of underlying habitats [24]. However, this may not be the case for territorial animals that conform to pre-emptive or ideal-dominant habitat distributions, where a habitat's intrinsic quality is better inferred from measures of fitness such as survival or reproductive success, implying the potential for source–sink population dynamics [23,24]. In addition, density-dependent habitat selection patterns may be more complex in territorial species [24,25]. For instance, when assessing temporal variation in habitat selection by territorial species, treatment of the available habitat distribution (i.e. the probability density function of all locations available to be selected over an area of interest; [26]) becomes complicated by the potential for exclusion [23,27] and/or despotism [22,28]. More specifically, the habitat distribution that is available to early arrivals is different from that available to later arrivals. Assuming early arrivals recognize and occupy the best available sites [23], the available habitat distribution for later arrivals will not include these sites until they are again vacated. By definition, evaluation of habitat selection in RSFs or HSFs depends on the ratio of used habitat to available habitat; if this ratio is ≠ 1, we conclude selection for (ratio greater than 1) or against (ratio less than 1) a defined habitat type [29,30]. The selection ratio is often assumed constant for relatively short-term studies, with inference occurring at the local population level. However, when temporal variation imposes changes in the used habitat distribution, the available habitat distribution, or both distributions, the resulting output can be difficult to interpret (e.g. [31]). When shifts in resource availability occur concurrently with changes in species density, attributing temporally varying habitat selection coefficients to the appropriate process can be challenging [14,32]. This is especially true when territoriality influences the habitat available to colonizing animals.

In this paper, we introduce a methodological framework to evaluate habitat selection by territorial animals in the presence of changing population density and environmental variation. To demonstrate our approach, we used a combination of long-term radio-collar and snow tracking data on grey wolves (*Canis lupus*) during a 19-year period of recolonization of the Upper Peninsula of Michigan, USA. Michigan wolves were an ideal case study population for this application, because they were

monitored intensively while their population recovered from less than 50 individuals to more than 600 [25,33]. Furthermore, patterns of wolf recovery and population expansion elsewhere in the Great Lakes region of the USA suggested density-dependent habitat selection trends, where the lowest risk sites (based on low human population density and agriculture) were the first to become occupied [34,35]. Once the highest quality sites were saturated, new wolf packs began to occupy lower quality sites (i.e. greater risk of human conflict), suggesting a pre-emptive population growth pattern [34,36]. Our application of a novel analytical approach to the Michigan wolf population, enabled us to (i) identify important predictors of wolf pack probability of site selection occurring within our study area, (ii) test for temporal variation in probability of selection, and (iii) explore density-dependent trends in probability of selection corresponding to the most important habitat predictors. For the latter objective, we focused on density-dependent habitat selection by explicitly accounting for functional responses that occurred due to long-term changes in habitat occupied associated with increasing regional wolf densities. Under the assumption of competitive exclusion, we applied resource selection probability function (RSPF) models [37,38] to each annual snapshot of wolf locations while constraining the available, unoccupied distribution primarily to areas not already occupied by existing wolf packs while accounting for potential territory overlap. We describe the foundation of this methodological workflow in the following paragraphs. Because many habitat selection studies are done using RSFs, as opposed to RSPFs, we first describe the relevance of density-dependent habitat selection within this context and demonstrate that the implementation of RSFs may be problematic when territorial exclusion is likely. We then describe our approach that combines data sources to implement an RSPF, thereby allowing identification of the unoccupied distribution.

## 1.1. Methodological background and framework

Assessment of density-dependent habitat selection requires spatially explicit information on species distribution and abundance that captures temporal variation at relevant scales [32]. These data are often gathered from long-term monitoring studies that include a population census or estimate of density that is repeated at a regular interval. Radio and GPS telemetry are perhaps the most broadly applied methods for monitoring species distribution across time [26]. The RSF [30], its equivalent HSF [5,32] and the RSPF [37,38] are widely established tools for analysing these data. These methods become increasingly powerful when combined with population information and/or indices of abundance [32,39]. In particular, the effect of density on habitat selection can reveal important insights about the realized as opposed to fundamental ecological niche [2,14,20]. However, density-dependent habitat selection has not yet received thorough exploration for territorial species. This may be partly due to the difficulty in accurately defining distributions of habitat availability [5,40,41].

We approached this problem by assuming that site occupancy and territoriality were the primary mechanisms imposing constraints on habitat availability as the wolf population increased. We first describe the problem's relevance to the commonly applied RSF, where used locations are contrasted with pseudo-absence, or background locations used to characterize availability. The RSF is commonly applied in telemetry studies, but may be more prone to violation of assumptions about the distribution of available habitat when considering territorial species. For those reasons, we describe the RSF and its potential for bias when applied to territorial species, before transitioning to our example case study that employs the RSPF where true availability is assumed to be known.

Following derivations of the use-availability likelihood in McDonald [42] and Aarts et al. [5], we define relative habitat use as the probability density function $f_u(X)$

$$f_u(X) = \frac{w(X)f_a(X)}{\int_E w(X)f_a(X)\mathrm{d}X} \tag{1.1}$$

where the available distribution $f_a(X)$ within a study area domain comprises environmental covariates $X$ in multi-dimensional environmental space $E$, and $w(X)$ is the RSF [5,41]. Equation (1.1) can be rearranged to show that $w(X)$ is proportional to the ratio of use to availability for the set of covariates $X$ [5]. Specifying habitat use as a weighted distribution [5,37] implies that changes in relative use depend on changes in availability, unless otherwise adjusted for in $w(X)$ [14,17]. This is an important consideration when applying an RSF to a population that exhibits territoriality, because the availability distribution may depend on occupancy by dominant or territorial individuals [28]. To estimate the exponential RSF $\hat{w}(X)$ and the set of habitat covariate effects $\hat{\beta}$, a generalized linear model (GLM) with binomial family and logit link function (i.e. logistic regression) can be used to

obtain the habitat selection parameters $\hat{\beta}$, conditional on the nuisance intercept parameter $\beta_0$ [42]. This is done by sampling used and available locations from the distribution of geographically available habitat (typically coded as 1's for used and 0's for pseudo-availability) under the following conditions: the availability sample $S_a$ is *iid* and represents all areas in $f_a(X)$ equally, the sampling domain $D$ is the same for the used sample $S_u$, $S_a$ and $S_u$ do not depend on each other and the exponential link is used to obtain the predicted values for $\hat{w}(X)$ [42]. It has been shown that this procedure is exponentially equivalent to the estimation of the relative intensity of an inhomogeneous spatial point process [43,44].

Taking these constraints into account, it becomes apparent that a density-dependent RSF for territorial animals can only be valid if the availability distribution is realistically constrained to the unoccupied landscape for a given individual or group. Achieving this would involve repetitively updating $D$ to match the conditions being observed by a given individual or group $i$ at a particular time $t$. In other words, traditional habitat selecting modelling approaches can be used, but care must be taken to appropriately model the dynamic boundaries of the available habitat distribution and subsequent used and available sampling domains. In general, an RSPF or occupancy model might be preferred for territorial species if data permit its estimation. An important distinction between the RSF and RSPF is that the RSPF relies on non-used locations that are assumed to be true absences [41,45]. This allows estimation of a legitimate probability (e.g. probability of use or occupancy) by applying an appropriate link function, such as logit or probit, to the linear predictor in a GLM [41,45]. To meet these criteria and implement the RSPF, we redefined the area available to each collared wolf pack in our study on an annual timestep, assuming it to be unoccupied by other wolf packs, with the use of census data (tracking) combined with telemetry observations. Our approach combined long-term occupancy patterns with radio-telemetry data to generate annual snapshots of area occupied, area available (i.e. unoccupied) and regionally varying wolf density. Thus, we were able to draw pack- and year-specific used and unused samples from a dynamic sampling domain accounting for changes in wolf distribution and density over time.

# 2. Methods

## 2.1. Overview

We used radio-collar data from grey wolves in Michigan, USA, to assess the effect of increasing wolf density on probability of habitat selection at the territory scale during a period of recolonization, 1994–2013 [25]. We expected wolves to recolonize based on the ideal despotic (IDD; [22]) or ideal pre-emptive distribution (IPD; [23]), where individuals either pre-emptively occupy highest quality sites [27] or claim territories based on competitive superiority [28]. By this expectation, highest quality sites would be selected first while abundance increased, until all the best sites were occupied. Packs would then compete for habitat and increasingly occupy marginal territories, potentially leading to source–sink dynamics and declines in vital rates and/or population growth rates [4,28,46,47]. We assumed that the distribution of habitat availability from an individual wolf pack's perspective varied depending on population abundance and local densities, such that areas occupied by existing packs became less attractive to new colonizers.

From a resource selection standpoint, this process would result in location and scale shifts in both the used and available habitat distributions, potentially leading to density-dependent habitat selection [14] as the used habitat distribution becomes increasingly dependent on the *per capita* available habitat distribution. Importantly, we note that density-dependent habitat selection in territorial species can take any form if the available habitat distribution is allowed to vary with occupancy. For example, preference or avoidance of a given habitat suitability predictor could increase if the rate of change in the available distribution exceeds the rate of change in the used distribution. Alternatively, preference or avoidance could become weaker or remain constant (figure 1; electronic supplementary material, appendix S1).

## 2.2. Data collection

Wolves were live-captured using foot-hold traps during spring and summer 1992–2013 as part of a larger Michigan Department of Natural Resources (MDNR) wolf monitoring programme [33]. Individuals were chemically immobilized (ketamine hydrochloride and xylazine, $100 \text{ mg ml}^{-1}$) using $0.11 \text{ mg kg}^{-1}$ ketamine hydrochloride and $2 \text{ mg kg}^{-1}$ xylazine and fitted with VHF radio-collars [48]. We located

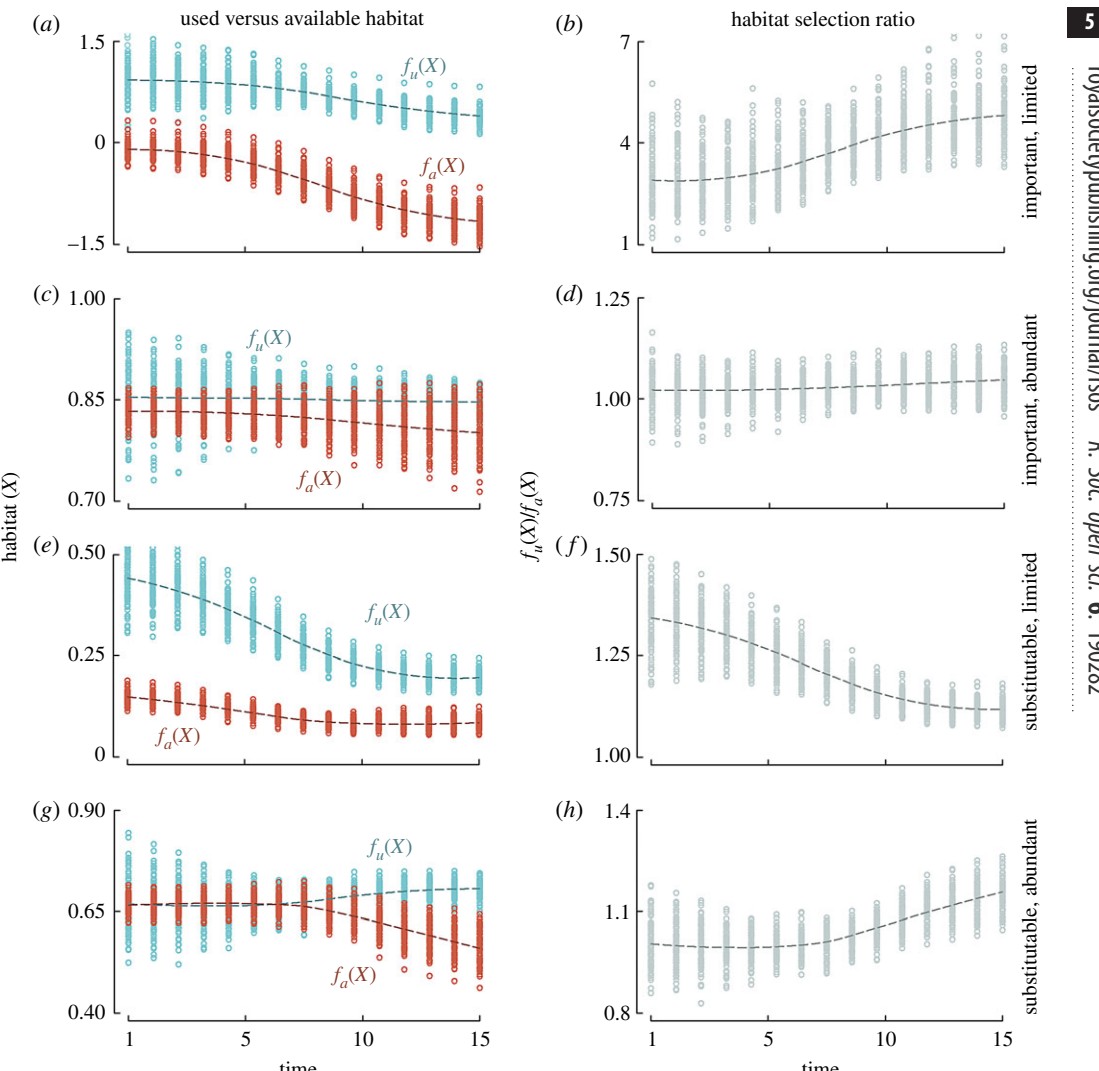

**Figure 1.** Results from simulations of used ($f_u(X)$) and available ($f_a(X)$) habitat (a,c,e,g) and corresponding selection ratios ($f_u(X)/f_a(X)$; b,d,f,h) under assumptions of strong territoriality and increasing occupancy over time. Four hypothetical scenarios were evaluated in simulations, including an important, limited habitat (a,b), abundant but important habitat (c,d), limited substitutable habitat (e,f) and more abundant substitute habitat (g,h). Results show that the change in the selection ratio is dependent on convergence or divergence between used and available habitat distributions as occupancy increases over time.

wolves by fixed-wing single-engine aircraft approximately one to two times per week. Further details regarding the telemetry study are described in [33,48,49].

We used data from Michigan DNR wolf track counts to estimate variation in wolf density over space and time. Track counts began in 1992 and continued throughout the duration of the study. The study site was divided into 21 units and all passable roads were surveyed during winter from trucks and snowmobiles [48]. Pack sizes and territory boundaries were established by intensive tracking efforts, with trackers using information from radio-collared wolves as well as recording all signs, such as territory markings, scat and individual sets of tracks [48]. An accuracy assessment of the ground tracking efforts was conducted during an independent study [49], which revealed a 4% average difference between the separate counts [33]. In 2007, the state adopted a geographically stratified sampling plan to reduce the cost and effort of the survey. A panel design was implemented to increase the precision of abundance estimates which ensured that some sampled units were counted during successive years [48,50].

## 2.3. Estimation of territory boundaries

Each wolf territory was established by a combination of radio-collar locations and track surveys. Following detection of a pack, territories were monitored either by aerial telemetry relocations from

at least one resident individual wolf or by repeatedly visiting the site via the annual tracking survey. This allowed us to document pack presence and territory persistence over the full course of the study. We delineated annual territory boundaries using the following framework: first, if at least 30 telemetry locations were available for a pack during a year (e.g. year = time $t$), we generated a unique territory home range for year $t$. If there were less than 30 locations for year $t$, but at least 30 locations were available over the course of a 2- or 3-year time period ($t-1$, $t$, $t+1$), we generated the territory home range using a 3-year moving window. For all other years that packs were known to be present at their site, we generated long-term average territories using either (A) locations from previous years, i.e. territories from previous steps, (B) a combination of telemetry locations and tracks from surveys, or (C) a minimum convex polygon based on track locations occurring over the full time series.

When telemetry locations were available ($n \geq 30$), we used a fixed kernel density estimator to create a utilization distribution (UD) for each pack territory during either year $t$ or the 3-year moving window. The kernel bandwidth was estimated using the 'plug-in' bandwidth estimator [51] after first removing outlying locations [52]. We defined the territory home range as the 95% volume isopleth from the UD. Home ranges and bandwidth estimators were analysed using packages 'adehabitatHR' and 'ks' in R 3.2.2 [51,53,54]. We plotted territory size as a function of method (A, B or C) to check for bias in size associated with differences in how boundaries were estimated (electronic supplementary material, appendix S2).

## 2.4. Annual wolf density

Packs were counted during track survey efforts. The entire study area was counted from 1995 to 2006. For survey units that were not surveyed every year starting in 2007, we assumed that packs persisted if they were detected the years directly before and after the year for which the count did not occur. We used the midpoint to extrapolate pack size in these cases. The last year included in the study was 2013, but surveys continued the following year, allowing us to use data from 2014 to make extrapolations. We created a longitudinal matrix with pack territory as the subject unit (rows) and year as the time unit (columns). For each year in the study, each pack was either detected, assumed present, or not detected, and pack size estimates were recorded in a related table. We summed rows of the matrix to estimate total wolves and compared results to MDNR's abundance estimates [55,56] to verify that our assumptions of occupancy and pack size were reasonable. The matrix was linked to a geodatabase with polygons for all territory home ranges estimated each year; all packs with counts greater than or equal to 2 were included in subsequent steps while lone individuals were assigned to remaining geographical space (i.e. area not occupied by an existing pack during year $t$). We converted pack sizes to density (wolves/1000 km$^2$) for each territory, and ultimately generated a smoothed surface for each year using a circular moving window with radius approximately equal to the median wolf dispersal distance (km), which was based on an exponential distribution with $\lambda = 1/55$ [57]. We performed this procedure to obtain a regional (rather than local) estimate of wolf density, thereby allowing local habitat selection effects to be conditioned on regional estimates that influenced regional habitat availabilities (e.g. characterizing density-dependent functional responses; [15,18]). Geoprocessing steps were completed in ArcMap 10.3 (Environmental Systems Research Institute, Inc., Redlands, CA, USA) using ArcPy for Python 2.7.2.

## 2.5. Uncertainty in territory occupancy

Because we could not assume perfect observation of pack territory boundaries and due to differences in methods, we performed a simulation to generate raster surfaces representing uncertainty in pack territory occupancy on an annual timestep. We applied these rasters as probabilistic resistance surfaces when drawing spatially balanced random points from available habitat distributions for each wolf pack-year (see *Sampling design and resource selection probability functions*). Specifically, for each year, we created 100 new, plausible versions of the territory boundary for each pack. We did this by randomly sampling a new area for each pack based on a normal distribution of territory areas. If telemetry data were used (method A or B) and multiple pack-years existed, we used the pack-specific standard deviation to sample new areas where the mean was either the current area (year = $t$; method A) or the long-term average area (method B). Otherwise, if telemetry data were inadequate across pack-years (method C), we drew the new area based on the mean and standard deviation of all pack UDs. Then, we expanded or shrunk each pack-year territory polygon based on the difference between the new

sampled area and the area of the baseline polygon. Once complete, we combined all pack territories for each year, converted them to binary rasters (900 m$^2$ pixel; territory versus non), and summed across the 100 rasters using raster cell statistics. The result, divided by 100, represented the probability that each pixel fell within a pack territory polygon, (i.e. the probability that cell $i$ was occupied by a wolf pack during year $t$). We generated this result for each year in the study. We automated geoprocessing steps using ArcPy (Python 2.7.2).

## 2.6. Landscape variables

To characterize habitat in the study area, we considered land cover and topographic characteristics (i.e. natural features), indices of prey availability, and measures of human infrastructure and density [34,58,59]. For natural features, we used 30 m digital elevation models (DEMs) to quantify topography, including measures of elevation, slope, topographic roughness and aspect. We used National Land Cover Data (NLCD) to quantify land cover characteristics, such as open areas (i.e. inverse of forested land), water/wetlands and edge habitat (interface between forested and open areas). Stream densities were also derived from Michigan's hydrography framework. Land cover characteristics were evaluated using the 1992, 2001, 2006 and 2011 products to represent any land cover change occurring during the time series. We considered several predictors assumed to be representative of prey availability on the landscape. Wolves primarily prey on white-tailed deer (*Odocoileus virginianus*) in this study region. However, most deer in the study region are obligate seasonal migrators due to heavy winter snowfall. Moreover, high concentrations of deer in winter are found in dense coniferous canopy cover, often consisting of eastern hemlock (*Tsuga canadensis*) and northern white cedar (*Thuja occidentalis*), which intercept large amounts of snowfall and provide important cover [60]. These deer wintering complexes (hereafter, DWCs) have been mapped by state biologists since the 1930s (see electronic supplementary material, appendix S3 for details). We estimated distance to DWC to represent winter prey availability using an exponential distance decay function, $\exp(-d/\alpha)$, where $d$ was the distance value at location $i$, and $\alpha$ was specified as the mean distance at all used locations (electronic supplementary material, appendix S3). We additionally quantified average annual snow depth and long-term average snow depth (average across all winters in the study).

Measures of human infrastructure, indices of human population density and proportion of public land were generated from NLCD, TIGER/Line roads files and the US Protected Areas Database (GAP Analysis program; see electronic supplementary material, appendix S3). The per cent impervious developed area data product was used as an index for human activity, agricultural land was extracted from NLCD land cover products, and minor roads and major highways were separated from each TIGER/Line file. Minor roads (primarily forest roads and snowmobile trails) were converted to density (km km$^{-2}$), while distance to highway was calculated using the exponential decay function. Protected areas were extracted from GAP products and comprised mainly land under state and federal ownership. We used moving window analyses to develop spatially explicit surfaces for each landscape feature considered. The moving window was necessary to calculate metrics representing the proportion of landscape or density of linear or point features, and we applied the moving window to all predictors to maintain a consistent scale of analysis. The size of the circular assessment window was set to 50.75 km$^2$, which was one-quarter of the mean wolf home range estimated during the study. This radial distance was chosen to incorporate finer-scale variability in habitat within territories, while also representing landscape-level habitat variability. Spatial variables were evaluated in ArcGIS 10.1 (Environmental Systems Research Institute, Inc., Redlands, CA, USA). We standardized each variable around its mean and standard deviation prior to model fitting. Full details on the data sources, development and representation of spatial landscape variables are provided in the electronic supplementary material, appendix S3. Spatial analysis of environmental features often produces correlated predictors that can interfere with model-fitting and interpretation of results [61]. To reduce the number of variables considered and avoid redundant predictors, we initially fit GLMs using penalized maximum likelihood to select a subset of predictors from the original set [62,63]. Details on preliminary model reduction are in electronic supplementary material, appendix S4.

## 2.7. Sampling design and resource selection probability functions

To characterize the used and unused distributions necessary for estimating an RSPF, we used a spatially balanced random sampling design [26], where habitat used by wolves was contrasted with unoccupied

habitat at the territory level (consistent with second-order scale of selection; [29]). Correspondingly, each year-specific pack territory boundary represented the sampling domain for used locations for individual $i$ and time $t$ ($S_u^{ti}$). The available, unoccupied sampling domain ($S_{un}^{ti}$) was specific to each individual and year as well, with random point distributions based on the probabilistic surfaces described in '*Uncertainty in territory occupancy*', where the inverse probability of occupancy was used to determine where random points fell within $S_{un}^{ti}$. We assumed that wolf dispersal followed an exponential distribution with a mean distance of 55 km ($\lambda = 1/55$; [57]). In this case, 95% of dispersal distances are less than 165 km, so we buffered 165 km from the centre of the individual's home range to set a maximum geographical range for $S_{un}^{ti}$. We restricted geographical unoccupied range for each pack-year by removing all areas known to be occupied at time $t$ from the buffered home range. This analysis was repeated for all packs and all years in the study. We estimated total per cent territory overlap on an annual basis by summing the areas of all pack territories, and then summing the areas of all geographical intersections between territories. We used the result to allow a proportion of unused locations to fall within other occupied territories each year, as informed by our data where territory overlap increased over time with increases in density. For each pack-year, we sampled uniformly within $S_u^{ti}$ and $S_{un}^{ti}$ as follows: we drew one random location to every 50 km$^2$ within $S_u^{ti}$, and one location to every 1000 km$^2$ within $S_{un}^{ti}$, where some random locations were drawn from other occupied territories, and other random locations were a spatially balanced sample based on the inverse probability of occupancy. Sampling in this fashion allowed the proportion of used to unused points to approximate the territory size and area available from each pack-year's perspective. This generally corresponded to approximately five used locations for an average territory size, with 'unused' locations depending on the proportion of overlap and the remaining unoccupied area. Lastly, all locations were updated with values from landscape variables, standardized estimates of wolf density and factors representing pack territory and biological year. We automated random location sampling (ArcPy with Python 2.7.2 and ArcMap 10.3).

We used a generalized linear mixed model (GLMM) framework to accommodate unbalanced subpopulations, repeated sampling of the same packs over multiple years and correlations that may otherwise exist among packs [64,65]. We used the binomial family of distributions, where used locations are coded 1 and unused locations coded 0, and the logit link was used to derive the RSPF, as opposed to an exponential link that would be required if the model were an RSF (see *Methodological background and framework*) [42]. To represent our time- and pack-specific sampling design, we modelled pack and year as random effects [64,65].

We implemented RSPF models with integrated nested Laplace approximation in R-INLA [66], a flexible computing environment for fitting a large variety of spatial and spatio-temporal models using a Bayesian hierarchical modelling framework [66,67]. We considered two final versions of the RSPF. First, we fit a latent Gaussian model akin to $Y \sim \mathbf{X}\boldsymbol{\beta} + f(t) + f(\mathbf{x},k)$, where $Y$ represents used versus unused observations, $\mathbf{X}$ is the matrix of landscape variables with associated regression parameters $\boldsymbol{\beta}$, $f(t)$ is a random *iid* intercept for year $t$ and $f(\mathbf{x},k)$ represents random coefficient effects for pack $k$. In the second model, for all landscape variables, we added a density-dependent interaction effect where the main effect of the variable was modelled as a function of wolf density [14]. By default, all regression parameters were assigned uninformative Gaussian priors. To estimate the RSPF from the final model, we included the mean of the marginal posterior density estimate for each $\beta$ in the RSPF model formula [37,45]

$$\hat{w}^*(\mathbf{X}) = \frac{\exp(\hat{\beta}_0 + \hat{\beta}_1 x_1 + \hat{\beta}_2 x_2 + \ldots + \hat{\beta}_k x_k)}{1 + \exp(\hat{\beta}_0 + \hat{\beta}_1 x_1 + \hat{\beta}_2 x_2 + \ldots + \hat{\beta}_k x_k)}. \tag{2.1}$$

Note that the above equation refers to the population-level estimate, which is not specific to any pack or year, and would be conditioned on the mean wolf density if interactions were present. We reported population-level effects for model covariates, corresponding to the averages across the full time series conditional on mean wolf density, including all packs and years. To demonstrate broad-scale spatial heterogeneity, density-dependence in probability of use, and temporal change, we also extrapolated fitted model values for the early (1995–2000), middle (2001–2006) and late (2007–2013) time periods of the study. These results were obtained by extracting pack- and year-specific fitted values from the model at each used/unused sample point, and generating spatially smoothed probability surfaces using empirical Bayesian kriging in ArcGIS 10.3. We evaluated the model fit by computing Pearson's correlation ($r$) between the mean predicted and observed values and assessing the model's ability to discriminate between occupied and unoccupied locations using the receiving operator characteristic

and area under the curve (AUC) statistic [68]. Finally, we performed leave-one-out cross-validation via the log-conditional predictive ordinate statistic [69,70]. R code and data to fit this model using R-INLA are available in the open Data Repository for the University of Minnesota (DRUM; https://doi.org/10.13020/s40h-fv72).

# 3. Results

## 3.1. Data attributes

During the course of the study, 371 individual wolves were captured and relocated by aerial surveys, and 30 091 locations were recorded. Track surveys identified 229 unique pack territories overall, with annual territory counts ranging from 33 (1995) to 102 (2006) before implementation of the stratified sample survey. The mean pack size during the study was approximately four wolves and increased over time, with annual means ranging from 2.74 (s.e. = 0.86) to 5.14 (s.e. = 3.40; electronic supplementary material, appendix S5 and table S5). Our estimates of wolf abundance, based on data and assumptions about pack occupancy and size, aligned closely with MDNR estimates ($R^2 = 0.98$; electronic supplementary material, appendix S5 and figure S5). The wolf population increased from an estimated 80 individuals (1995) to 687 (2011) before evidently stabilizing [55,56]. The overall population growth rate during this time declined from $r = 0.16$ (1995–1996) to $r = -0.01$ (2009–2010, 2011–2013), and appeared consistent with density-dependent logistic growth [71]. Estimates of the mean wolf density (minimum $N$/total study area) increased from approximately 1.86/1000 km$^2$ in 1995 to more than 15/1000 km$^2$ in 2011–2013 (electronic supplementary material, table S5) and were geographically variable (figure 2). Wolves apparently recolonized the majority of suitable habitat during the study, and the estimated proportion of territory overlap increased concurrently with increasing wolf density (figure 3a), from less than 10% (1995–1997) to approximately 48% (2013). Figure 3 demonstrates potential changes in selection ratios when used and available distributions of important habitat covariates change corresponding to changes in wolf density. Tracking and radio telemetry revealed 137 individual packs occupying approximately 63% of the study area by 2011 (electronic supplementary material, table S5). The average pack territory area estimated from telemetry was 283.10 km$^2$ (s.e. = 171.41) and 282.36 km$^2$ (s.e. = 158.33) using all data types (telemetry and track survey data; electronic supplementary material, appendix S2 and figure S2).

## 3.2. Variable selection

Penalized maximum-likelihood model reduction procedures resulted in dropping 7 of the initial 16 candidate predictors of wolf habitat: % open cover types, % water/wetlands, distance to highway, % agriculture, topographic radiation aspect index, topographic roughness and annual average snow depth. Dropping predictors indicated that they did not contribute substantially to predictions of wolf territory selection probability, or that other correlated predictors had stronger predictive capacity. Predictors retained in the subsequent modelling included proximity to DWC (winter prey index), stream density, slope, elevation, long-term average snow depth, density of forested-open habitat edge, minor road density, % impervious developed and % public land (table 1). Marginal posterior distributions for habitat covariate effects in the final model (interacting with wolf density) indicated that wolf territory probability was driven primarily by human influence (% impervious developed), winter prey availability (proximity to DWC), topography (elevation and slope), forested versus open habitat edges, minor road density, long-term average snow depth and stream density (table 1). Many of these effects were dependent on variation in wolf density (posterior 95% CIs for interactions not overlapping zero). Posterior CIs for density-dependent interactions were relevant (i.e. 95% CI did not include zero) for many effects even when the main effect, conditioned on the mean wolf density, was not a strong predictor of habitat selection (e.g. stream density, forested-open edge, elevation; table 1 and figure 4).

## 3.3. Model fit and parameter estimates

Diagnostics from our final model indicated strong model fit ($r = 0.835$, AUC = 0.969). The density-dependent model out-performed the model without density-dependent interactions (WAIC = 15 609

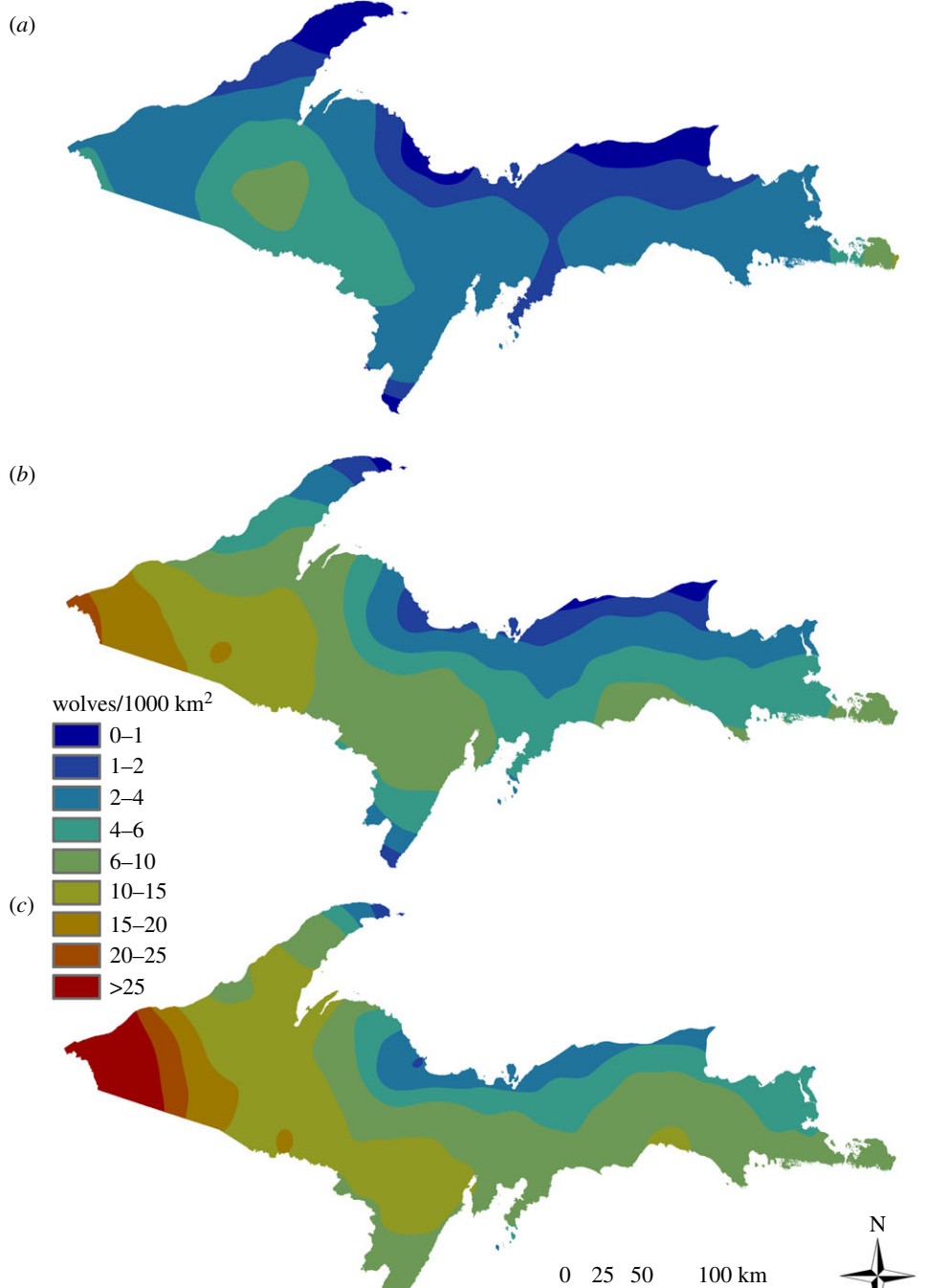

**Figure 2.** Spatially explicit smoothed estimates of wolf density from early recovery (1995–2000) (*a*) to late (2007–2013) (*c*) in Michigan, USA. Wolf density estimates were generated from winter tracking data and radio telemetry, with pack territories and sizes being monitored annually during the study. Pack sizes and territory locations were converted to density (wolves/1000 km$^2$) for each territory and smoothed using a circular moving window with radius approximately equal to the median wolf dispersal distance (km), which was based on an exponential distribution with $\lambda = 1/55$. Smoothed surfaces were used to represent regional wolf density, and thereby explore habitat functional responses to changing habitat availability; estimates were averaged across time periods to create snapshots for 1995–2000 (*a*), 2001–2006 (*b*) and 2007–2013 (*c*).

versus WAIC = 15 929; *d*WAIC = 320), although model diagnostics suggested both models reliably predicted the observed data (no density-dependent interactions: $r = 0.830$, AUC = 0.967). Hence, allowing pack-specific coefficients resulted in strong performance for both models. Model estimates from the final model suggested evidence for density-dependent habitat effects (table 1 and figure 4). Noting that the model estimates were conditioned on the mean wolf density (i.e. centred), the effects

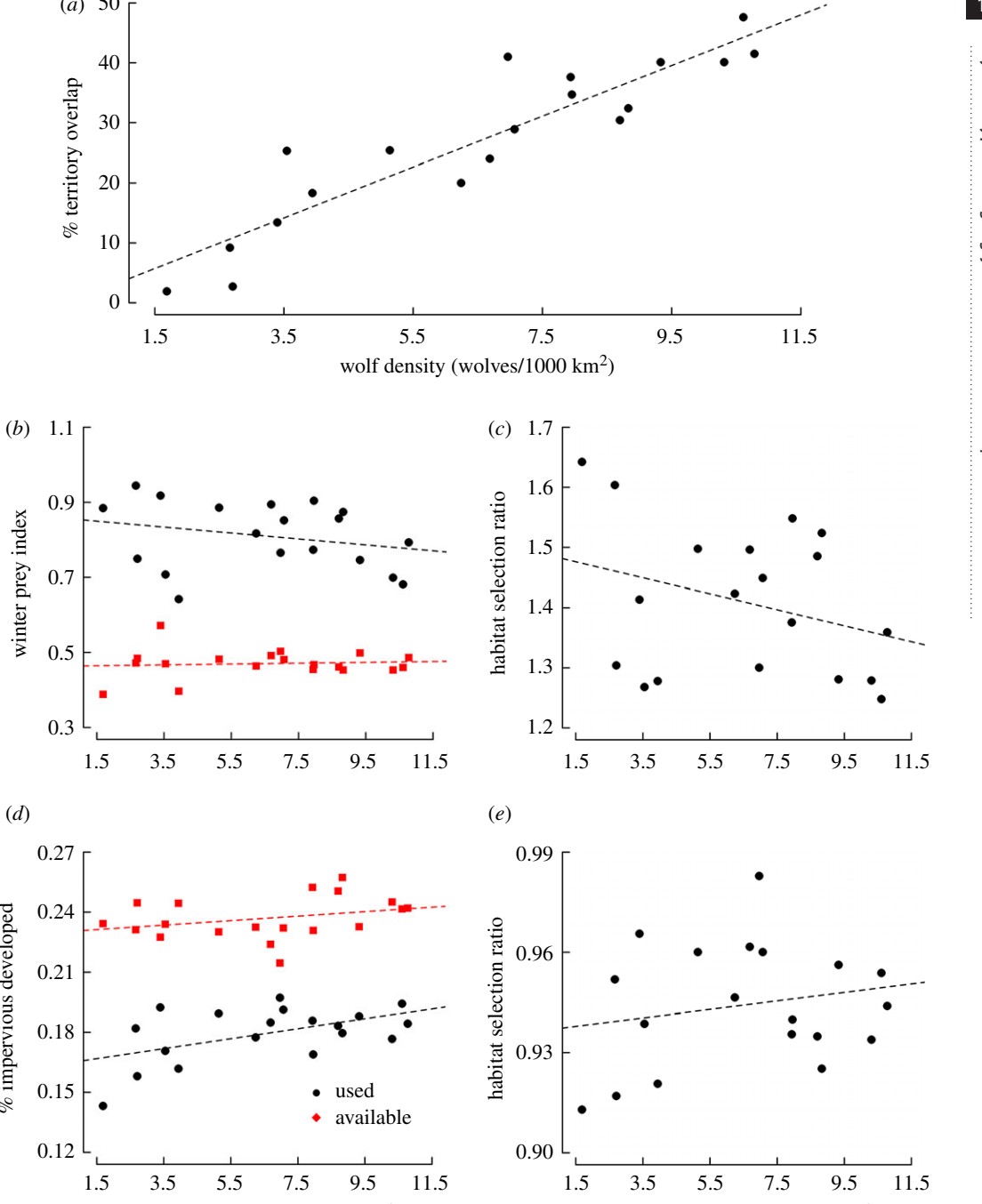

**Figure 3.** Estimation of % territory overlap (*a*), and comparisons of means for distributions of used and available winter prey availability and proportion of impervious developed area (*b*,*d*), along with ratios of used to available habitat (*c*,*e*) corresponding to changes in wolf density for wolves in Michigan, USA, 1995–2013.

of elevation and stream density were positive at low wolf densities but became negative at high wolf densities ($\hat{\beta}_{ELEV \times WDENS} = -0.325\,[-0.401, -0.250]$, $\hat{\beta}_{STREAM \times WDENS} = -0.137\,[-0.207, -0.068]$; figure 4), while the effect of winter prey decreased with increasing density (shallower slope; $\hat{\beta}_{DDWC \times WDENS} = -0.110\,[-0.170, -0.050]$). In addition, the negative effect of slope decreased (became less negative; $\hat{\beta}_{SLOPE \times WDENS} = 0.161\,[0.081, 0.242]$), and the effect of habitat edge switched from negative at low densities to positive with increasing density ($\hat{\beta}_{EDGE \times WDENS} = 0.205\,[0.108, 0.303]$). All other density-dependent terms had effects overlapping zero (table 1). Relative effect sizes are more easily interpreted graphically, with wolf pack occurrence conditioned on an initial probability of 0.5 (figure 4). For example, the change in the slope for winter prey is evident in figure 4*c*, with the overall effect of winter prey remains positive even at the highest wolf density. By contrast, the sign of the main effects for stream density and

**Table 1.** Posterior marginal distributions for predictors of wolf territory occurrence probability in Michigan, USA, 1995–2013. Predictors were modelled as a function of wolf density, which varied both spatially and temporally. Results of main effects are conditioned on the mean wolf density, as all fixed effects were centred during model fitting. Models were fit using random effects for time and pack territory, thus accounting for repeated sampling of occurrence over time. Posterior distributions for each parameter were estimated using integrated nested Laplace approximation in R (R-INLA), and reported effects correspond to the population-level effect while accounting for pack-level variation, ordered by relative effect size $\mathrm{abs}(\hat{\beta}/\mathrm{s.e.}(\hat{\beta}))$, where predictors with 95% CI not overlapping zero are in italics.

| parameter | mean | s.e. | 2.5th percentile | 97.5th percentile | mode | mean/s.e. |
|---|---|---|---|---|---|---|
| *intercept* | −8.089 | 0.272 | −8.632 | −7.63 | −8.082 | −29.768 |
| *wolf density* | 0.724 | 0.050 | 0.636 | 0.843 | 0.716 | 14.377 |
| *% impervious* | −1.203 | 0.148 | −1.503 | −0.920 | −1.197 | −8.132 |
| *winter prey index* | 0.287 | 0.067 | 0.155 | 0.418 | 0.287 | 4.286 |
| *slope* | −0.509 | 0.129 | −0.763 | −0.256 | −0.509 | −3.940 |
| *minor road density* | 0.216 | 0.074 | 0.071 | 0.361 | 0.217 | 2.930 |
| *avg. snow depth* | 0.601 | 0.300 | 0.012 | 1.191 | 0.600 | 2.002 |
| elevation | 0.385 | 0.280 | −0.165 | 0.933 | 0.385 | 1.377 |
| % public land | 0.068 | 0.156 | −0.238 | 0.374 | 0.068 | 0.437 |
| stream density | −0.009 | 0.084 | −0.175 | 0.157 | −0.008 | −0.102 |
| forested-open edge density | −0.004 | 0.110 | −0.220 | 0.211 | −0.004 | −0.038 |
| Posterior distribution of interaction (**x** × wolf density) | | | | | | |
| *elevation* | −0.325 | 0.039 | −0.401 | −0.250 | −0.325 | −8.426 |
| *forested-open edge density* | 0.205 | 0.050 | 0.108 | 0.303 | 0.205 | 4.141 |
| *slope* | 0.161 | 0.041 | 0.081 | 0.242 | 0.161 | 3.934 |
| *stream density* | −0.137 | 0.036 | −0.207 | −0.068 | −0.137 | −3.871 |
| *winter prey index* | −0.110 | 0.031 | −0.170 | −0.050 | −0.110 | −3.574 |
| % public land | −0.058 | 0.031 | −0.119 | 0.004 | −0.058 | −1.831 |
| % impervious | −0.109 | 0.075 | −0.256 | 0.038 | −0.108 | −1.453 |
| avg. snow depth | 0.058 | 0.050 | −0.041 | 0.157 | 0.059 | 1.152 |
| minor road density | 0.014 | 0.040 | −0.065 | 0.093 | 0.014 | 0.349 |

for elevation were positive at low wolf density and negative at high wolf density, indicating density-dependent switching (figure 4*a,f*). The opposite transition was observed for the forested-open edge effect (figure 4*d*). The negative effect of slope was stronger at low densities than at high densities, but remained a negative effect at all densities (figure 4*e*).

## 3.4. Mapping habitat selection

Our RSPF predicted density-dependent changes over time, as wolf pack occupancy expanded from the beginning of the study (1995) to the end (2013). Modelled probability of pack occurrences captured this dynamic (figure 5).

# 4. Discussion

Understanding density-dependent habitat selection for territorial species is valuable for multiple reasons. Changes in density or abundance may alter the response curves of predictors in a habitat model, particularly if the population has undergone long-term growth or decline during a study's time series.

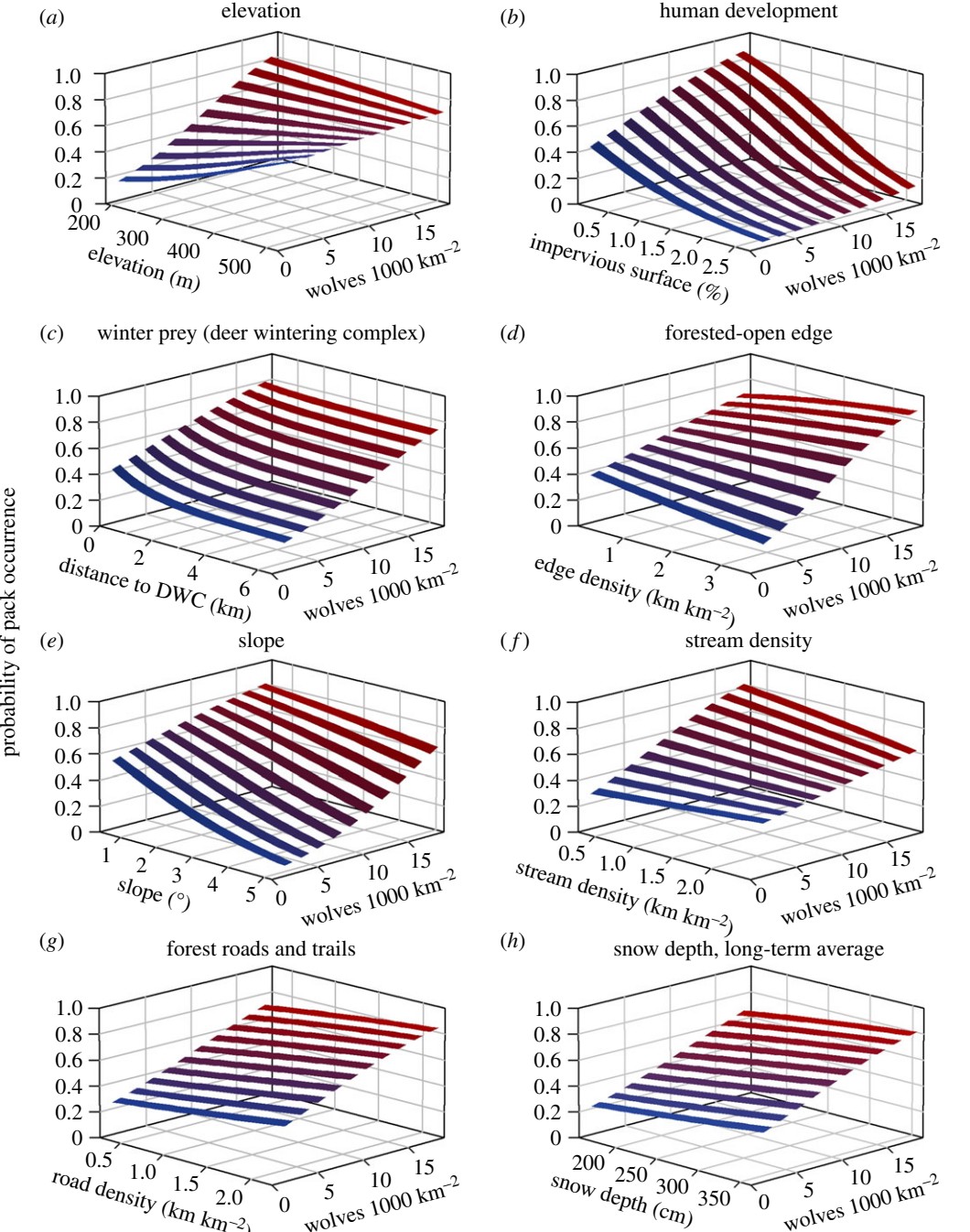

**Figure 4.** Probability of wolf pack occurrence from a RSPF for wolves in Michigan, USA. The RSPF was generated based on a GLMM framework using the R package 'INLA' with coefficient estimates generated conditional on specific wolf packs and years, and predictors of wolf pack occurrence were fit as interactions with wolf density, which varied spatially and temporally over time. Density-dependent habitat functional responses were observed with respect to multiple predictors, as indicated by a shallower slope with increasing density (c,e), switching from negative to positive (d) or positive to negative slope (a,f), or a steeper slope with increasing density (b).

Furthermore, predictions of the ecological niche based on current conditions may be unreliable because populations are not likely to be at equilibrium with their environmental surroundings at any given time, though often assumed to be [72]. In such cases, habitat quality may be reflected by habitat selection or occupancy patterns only at low population densities [19], especially under assumptions of theoretical ideal despotic or pre-emptive habitat distributions. Inferences from models that include density data while accounting for corresponding changes in habitat availability are also more likely to reflect population dynamics that are of interest. Variation in the population growth rate is likely to be

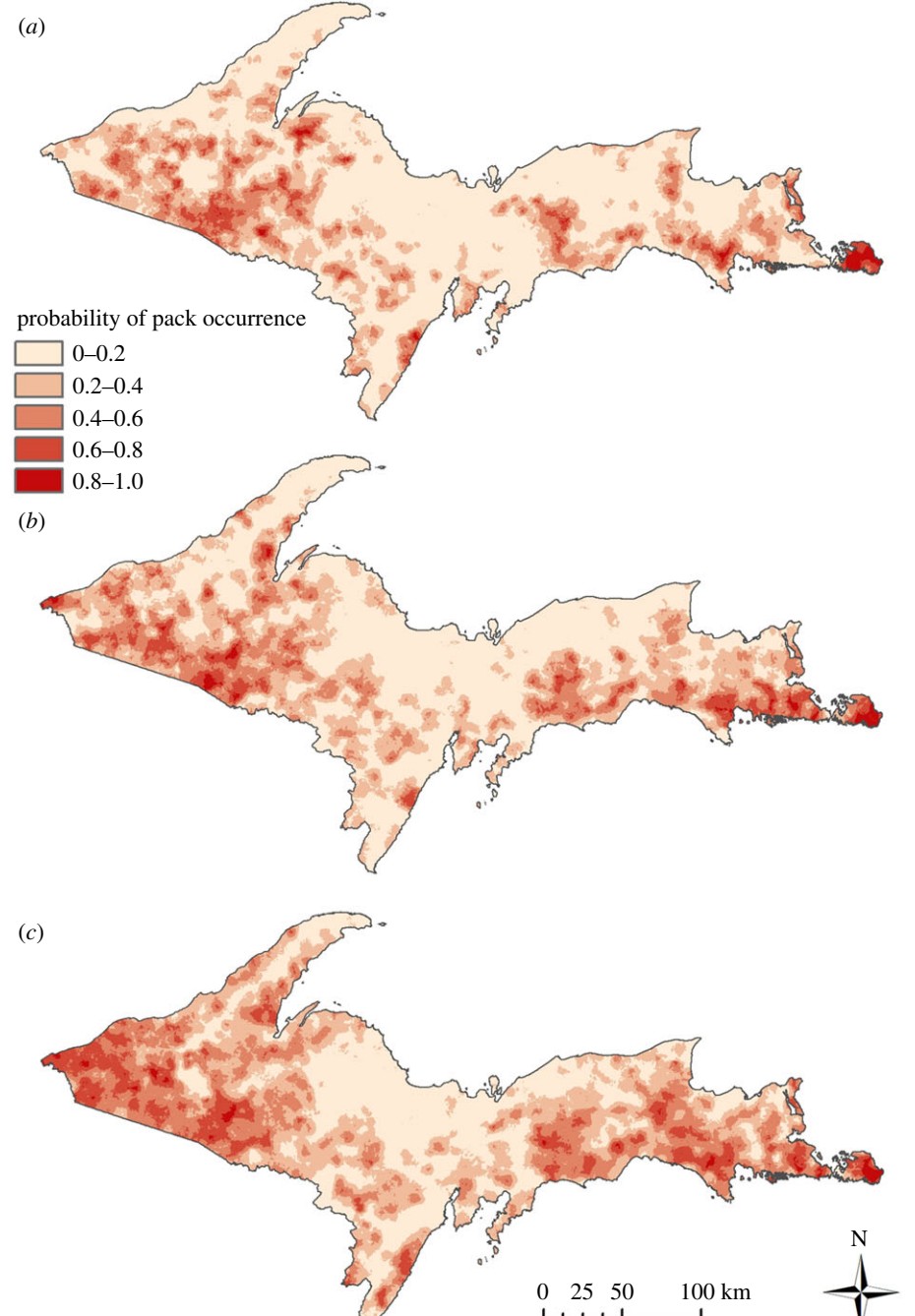

**Figure 5.** Model-fitted probability of wolf pack occurrence for three time periods during wolf recovery in Michigan, USA ((a) 1995–2000; (b) 2001–2006; (c) 2007–2013). Fitted probabilities were generated from an RSPF which was developed from a GLMM using the R package 'INLA' with coefficient estimates generated conditional on specific wolf packs and years and density-dependent coefficient interactions.

partially dependent on habitat selection [24,32], with positive average growth rates typically associated with higher quality habitat [2,4]. Furthermore, detecting behavioural changes associated with density-dependent habitat selection can reveal much about the limiting nature of important habitat predictors (figure 1; electronic supplementary material, appendix S1). For example, the distribution of used habitat typically will shift towards lower quality habitat as population density increases and quality sites become saturated; however, the available distribution (i.e. what remains and is not used) may also shift if quality habitat is limited (figure 1a). If the decline in the available distribution is steeper than that of the used distribution, then the strength of habitat selection *increases* (i.e. rather than decreases; figure 1b), indicating the finite nature of the given habitat predictor while demonstrating its importance to the species.

Understanding the influence of density-dependent mechanisms on habitat selection or occupancy patterns is especially critical for territorial and social carnivores such as wolves, particularly in areas where recolonization or range expansion is occurring. Territorial species should increase their use of suboptimal habitat types as density increases [22–24]. This has been observed to some extent in Great Lakes wolves [34–36], but focused evaluation has been rare. We demonstrated that wolves in Michigan, USA, exhibited density-dependent habitat selection patterns with the use of an RSPF that incorporated density interactions with influential habitat covariates. Our data and analysis supported the idea that wolves' distribution of used habitat shifted towards potentially lower quality habitat characteristics at the population level, but that distributions of unused habitat showed minimal change when considering increased territory overlap occurring in responses to increasing density as the best habitats became saturated (figure 3). Specifically, it appeared that new colonizing wolves became less likely to avoid existing territory boundaries as the population expanded, perhaps preferring to risk territorial disputes over settling in lower suitability habitats. Such a phenomenon could lead to increased competition for resources with possible density-dependent influences on reproductive success, dispersal and survival. This finding appears to be consistent with other studies in the Greater Yellowstone Ecosystem and the Great Lakes regions, where interspecific conflict increased with density and led to reduced survival in an unexploited population [46,73], while increasing density corresponded with riskier behaviours, increased human conflict and mortality risk in populations with greater exposure to lethal management, harvest and poaching [50,74]. Nonetheless, a decline in the effects of winter prey availability and stream densities (figure 4 and table 1) also demonstrated potential pre-emptive habitat selection behaviour.

Wolves' probability of selection response to areas with greater prey availability declined as density increased (table 1 and figure 4c). Similarly, positive responses to greater stream densities at low wolf density switched to negative as wolf density increased (figure 4f). Wolves initially occupied sites with higher relative elevation overall, but this effect declined with increased wolf density (figure 4a), while the opposite transition occurred in response to edgy habitat types (forested-open edge density; figure 4d). We assumed that density-dependence was primarily responsible for functional responses in habitat probability of selection. For instance, areas with high stream densities probably comprised better hunting habitats, as they implicate travel corridors and increased prey encounter rates [75], as well as potential alternative prey sources such as beavers (Castor canadensis; [76]). Areas of high stream density occurred largely in the western section of the study area, and were saturated earlier in the study, possibly contributing to the density-dependent response to stream density. Alternatively, some of the changes we observed may have been related to a general trend in wolf expansion from west to east across the study region. In our study system, changes in density varied spatially and on multiple time scales which complicated interpretation [25]. For example, wolf density was higher in the western half of our study area for much of the study and increased more substantially in the eastern half later in the study. This may have explained density-dependent responses to topographic variables, such as a declining response to elevation (higher elevations in the western half of the study region), and an increasing response to edge density (more open habitat types in the eastern half of the study region).

With respect to human impact, wolves avoided developed areas, but the modelled effect was not strongly density-dependent (table 1). Though wolves' use of developed area did increase slightly with increasing density over time, the same was true for the habitat classified as unused, so the ratio of used to available unused habitats was relatively constant (figure 3). Because anthropogenic development tends to increase over time, a density-dependent functional response could potentially be confounded by such changes. Despite avoidance of more developed areas, wolves responded positively to density of minor roads (e.g. forest roads and snowmobile trails), and this effect was generally constant with density (table 1 and figure 4g). Networks of linear features, such as roads and trails, may facilitate the ease of movement and increased encounters with prey, while being associated with forest cutblocks that promote higher deer densities in summer [77]. Forest roads were prevalent throughout the study area, so it is perhaps unsurprising that density-dependent changes were not indicated by our model.

Site dependence, involving occupancy or saturation of highest quality sites [27], may be a useful alternative mechanism to consider when modelling population density effects on habitat selection and distribution for territorial species. Although not independent of density-dependence, site dependence emphasizes the availability and quality of sites as a primary regulating factor [27]. Under pre-emptive habitat selection and a site-dependent theoretical model, sites with greatest suitability are selected above those of lesser suitability until the greatest suitability sites are all occupied [23,24,27]. Negative feedback in demographic rates occurs when lower quality sites are increasingly selected [27].

The ideal pre-emptive habitat distribution is likely for wolves, and localized increases in density occurred more rapidly in certain areas with greater prey availability and lower human influence in our study area [25]. Furthermore, density-dependence appeared to operate on a shorter time scale in these areas, and was lagged in areas with fewer high suitability sites. Under site-dependent regulation, density may be a poor predictor of large-scale changes in geographical availability of resources or suitable habitat sites. Wolf packs consistently occupy the same territory regardless of pack size, though pack size can factor into the ability of packs to defend their territory [73]. In addition, density can increase rapidly in an area of high suitability and crowding can occur due to territory overlap, while landscape-level occupancy stays relatively constant. As a result, functional responses in habitat selection may be alternatively explained by landscape-level occupancy as opposed to density when considering models that compare used with available habitat metrics for territorial species.

Spatio-temporal change in habitat model effects is expected when data are collected over a time series [5,32]. Accurately modelling a dynamic availability (or unoccupied) distribution is therefore critical, because density-dependent habitat functional responses are likely [5,17], especially in territorial species undergoing population change. Density-dependent habitat selection dynamics in ideal-free consumers have received some recognition [14,19], but less attention has been given to the case where changes in territory occupancy influence the used and available habitat distributions for species that exhibit strong territoriality, and thus theoretically follow an IDD or IPD [24,78]. Pre-emptive or territorial site selection patterns are important, because they implicate a discrepancy in fitness potential between habitats that may otherwise go undetected [79]. As such, it may be important to explore pack or individual-level heterogeneity with respect to performance (i.e. reproductive success or survival) in conjunction with habitat preference [80].

We used random effects within our model framework to assess pack-specific variation with respect to habitat influences on territory occurrence patterns. Assuming existing territories had reduced availability to new colonizers, we were able to redefine geographical habitat availability on an annual time step. As such, temporal changes in habitat coefficients associated with changes in occupied distribution were accounted for. Random effects have an additional benefit that is not often taken advantage of in habitat modelling efforts, namely the opportunity to explore individual- or group-level heterogeneity [65]. Random coefficients (also referred to as random slopes) defined on the individual or group with respect to one or more covariates capture deviations from the overall population-level mean. These deviations can then be linked to performance-based metrics such as survival or reproduction [80–82]. Although this was not our primary objective, our modelling framework presented the opportunity to explore differences in probability of selection in relation to habitat characteristics among wolf packs, so we briefly present an example. In a *post hoc* analysis, we compared average annual pack sizes for packs that exhibited greater positive response to the prey availability index ($\beta$ coefficient > 75th percentile value) with those with a weaker response ($\beta$ coefficient < 25th percentile value). Pack size increased as the coefficient for prey increased ($t = 3.99$, $p < 0.001$), and in a non-parametric Mann–Whitney distribution test, packs with weaker selection had lower pack sizes (2–9 wolves, $\bar{x} = 2.99$ wolves) than those with stronger selection (2–13 wolves, $\bar{x} = 4.19$ wolves; $W = 1102$, $p = 0.001$). This result may indicate a fitness consequence associated with density-dependent habitat selection, where packs that recognize and occupy territories with greater prey availability may be able to achieve greater productivity, i.e. through enhanced reproduction, survival and recruitment. However, it is again important to recognize that selection or probability of occupancy patterns under the constraints of territorial behaviour may be driven more by local or regional conditions (e.g. availability of prey in unoccupied areas) than an individual or group choice. The latter result is supported by our modelled evidence of a decline in use of greater prey availability with increases in wolf density.

## 5. Conclusion

Territorial animals should exhibit habitat selection patterns that are fundamentally different from those of more gregarious species. The effect of increasing occupancy and density over time means that what is perceived to be available to early colonizers may be vastly different from later occupants. Accounting for density- or occupancy-dependent habitat selection at the landscape scale is critical for identifying and understanding key limiting habitat factors and their relative availabilities. In Michigan wolves, we demonstrated density-dependent changes in the habitat attributes influencing territory site selection,

that were likely influenced by shifting perceptions of habitat availability as much of the landscape became occupied. This is an important example of a habitat selection functional response driven by colonization, density-dependence and long-term changes in occupancy.

Ethics. All capture and handling protocols were approved by Michigan Department of Natural Resources, State Wildlife Veterinarian. We followed American Society of Mammalogists guidelines for use of wild animals in research [83]. Capture and handling of wolves was permitted under Michigan's §6 Cooperative Agreement with the United States Fish and Wildlife Service in accordance with 50 CFR 17.21.

Data accessibility. Code and data to perform model reduction and fit the hierarchical wolf habitat selection RSPFs using the R-INLA library are archived in DRUM (Data Repository for the University of Minnesota) and are currently accessible at the following location: https://doi.org/10.13020/s40h-fv72 [84].

Authors' contributions. S.T.O., D.E.B. and J.K.B. conceived ideas, compiled long-term datasets, designed methodology and contributed critically to initial drafts and revisions; S.T.O. analysed the data and led writing of the manuscript; all authors gave final approval for publication.

Competing interests. We declare that we have no competing interests with respect to this study.

Funding. This work was primarily supported by funding from the Michigan Department of Natural Resources (Michigan Technological University grant no. 751B4300037) and additionally funded by Federal Aid in the Wildlife Restoration Act under Pittman-Robertson (project no. W-147-R), the National Science Foundation (grants to J.K.B.; NSF ID no. 1545611, NSF ID no. 1556676), the DeVlieg Foundation, and the Ecosystem Science Center within the School of Forest Resources and Environmental Science at Michigan Technological University, Michigan, USA.

Acknowledgements. We thank Erin Largent, Robert Doepker, Steven Carson, Brian Roell and Chris Webster for assisting with data needs during analysis. Mike Haen, Brad Johnson, Donald Lonsway, Jeff Lukowski and Kristie Sitar assisted with capturing and radio-collaring wolves. Pilots Neil Harris, Dean Minett and Gordon Zuehlke collected collared wolf relocation data. Anna Nisi and Emily Fifelski assisted with data processing. Additional thanks go to two anonymous reviewers for helping us improve the manuscript.

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
