## [Reviewer comments · Royal Society Open Science]

Review History

RSOS-190282.R0 (Original submission)

Review form: Reviewer 1

Is the manuscript scientifically sound in its present form?

Yes

Are the interpretations and conclusions justified by the results?

Yes

Is the language acceptable?

Yes

Is it clear how to access all supporting data?

Yes

Do you have any ethical concerns with this paper?

No

Have you any concerns about statistical analyses in this paper?

I do not feel qualified to assess the statistics

Recommendation?

Accept with minor revision (please list in comments)

Comments to the Author(s)

Review of RSOS-190282

O'Neil et al. Territorial landscapes: incorporating density-dependence into wolf resource selection study designs

This manuscript reviews the complexity that density dependence may introduce into both the understanding of habitat selection and attempts to quantify it. The authors then take a use/available approach and use a long-term telemetry data set for wolves in Michigan to explore the nature of density dependence on the drivers of habitat selection over a roughly 20 year period.

This paper is an interesting treatment of an issue that has important theoretical, practical, and policy dimensions – both in terms of what it adds to the ecologist's toolbox (intellectually and practically) and for what it helps us to understand about the expansion of wolves in the Great Lakes region. I think this is an important and worthy piece of research. It is novel and creative and complete in terms of describing the problem and the analytical approach and the reporting of results.

The manuscript is a bit schizophrenic in that it tries to tell two stories concurrently, one about theoretical issue of identifying and density dependence in habitat selection (either generally or in the case of territorial animals) and another about the pattern of wolf expansion in Michigan. I think it can do both because they are complimentary issues (as the authors acknowledge). The ability to tell two stories doesn't require a strict rule about how much content should be allocated to each but the manuscript feels the weakest where one seems story seems missing or not treated as completely as it could be.

Along those lines, conspicuous by its absence is the recent and earlier work done on density dependence in Wisconsin (full disclosure, I was a collaborator on that work) – after all the scientists in Wisconsin were studying the western half of the same population during the same time period. Van Deelen (2009) and Mladenof et al (2009) both speculated about the pattern of settlement and growth stemming from a phenomenon of wolves saturating high-quality habitat first. Indeed Mladenof's series of papers re-doing the habitat model of the Great Lakes looks like an illustration of the pattern that the authors analyze so compellingly here. Similarly, the individual-based model that Stenglein et al (Ecological Modeling) published duplicated the growth pattern seen in the Midwest by making the virtual wolves behave in a habitat quality based pre-emptive pattern. Other papers by Stenglein et al demonstrate density dependence in components of growth over time, that also varied spatially.

Specific comments:

Abstract: The abstract is generally very good. Length is about right given content and the nut of the analysis is well-handled. My quibble is over the concluding sentence. It concludes with a vague statement of territoriality pitched at analysts. I would add a sentence about what it tells us about the biology of wolf recovery in Michigan.

Introduction: The introduction was similarly well done. The problem of failing to deal with the dynamic nature of habitat selection in analyzing for habitat selection is laid out in a compelling

way. It could use a paragraph addressing the expansion of Michigan wolves and why habitat selection could be important in that context.

Ln 74: What's "habitat matching"? Is this another term for selection?

Methods: To be honest, the statistical modelling is a bit opaque to me but what I could piece together suggested a thoughtful, if sophisticated approach that tried to deal honestly and realistically with the data and analytical techniques that could be brought to bear. Appendix 2 was very useful in helping to understand the datasets and the treatments that those datasets received to become drivers of use and availability.

Ln 276,277: The sentence structure seems to contrast "territory" with "pack". Aren't these the same thing? Please clarify.

Ln 286,287: Why the imbalance here? Five random locations for use but 25 for available? With small samples could differences between use and availability stem from small sample phenomena?

Results: Well done. Model fit was remarkable!

Discussion: The discussion was also well done although I would recommend that the authors weave a bit more about what they learned about wolf recovery in Michigan as they discuss the theoretical points.

Ln: 444-445 Isn't site dependence a version of density dependence. Perhaps a definition to make things more explicit?

Table 1: Good.

Table 2: I presume the use of bolded text is acceptable for the journal... Otherwise good.

Figure 1: Good.

Figure 2: Good.

Figure 3: Good.

Figure 4: Good.

Figure 5 Good.

Figure 6 Good.

Review form: Reviewer 2 (Tal Avgar)

Is the manuscript scientifically sound in its present form?

Yes

Are the interpretations and conclusions justified by the results?

Yes

Is the language acceptable?

Yes

Is it clear how to access all supporting data?

Yes

Do you have any ethical concerns with this paper?

No

Have you any concerns about statistical analyses in this paper?

Yes

Recommendation?

Accept with minor revision (please list in comments)

Comments to the Author(s)

The MS titled “Territorial landscapes: incorporating density-dependence into wolf resource selection study designs” described an in-depth investigation of a habitat-selection probability function in the presence of territoriality. Overall I found the MS well written and providing the most thorough treatment of density-dependent habitat-selection analysis I know of; it will make a valuable contribution to the scientific literature.

That said, I do have two major(ish) methodological concerns. First, home-range (HR) delineations were done using at least three different approaches: an annual KDE (with variable number of locations as long as >30), 3-year or ‘long-term’ KDE (with variable number of locations as long as >30), or an MCP. Whereas none of these is ‘right’ or ‘wrong’, they may result in very different estimates and have different level of sensitivity to the number of locations and the period over which they were taken. I would be reassuring if the authors could provide a plot comparing HR size as function of these different methods, and a plot showing the relative proportion of each methods used as function of population density. My second, somewhat related, concern is the choice of five random points from the used domain and 25 random locations from the unused domain, regardless of the size of these domains. This means that the size of each home range is completely ignored in the analysis, which may be misleading as HR size might also shift with pack-size, number of neighboring packs, and time (due to familiarity and seniority effects). This also means that the intercept of the HSPF (Eq. 2) is biased. In theory, this intercept would be unbiased only under sampling design in which the numbers of used and unused points are proportional to the area of the respective used and unused domains. In practice, the main effect of population density may, to some degree, correct this bias, but I find it make interpretation extremely difficult. From my perspective, it would make more sense to sample point uniformly from the respective domains and not include a main effect of density. This would also mean that integrating the resulting HSPF over the study area for any specific year should yield (approximately) the total number of wolves occupying the study area in that year, which should provide a nice validation of the model.

Itemized comments:

- Lines 72-75: I think this could use some clarification. Under simple IFD with discrete habitat patches, occupancy is expected to be density dependent (poorer patches get occupied as density increases and they become relatively profitable), but among occupied patches, relative density should reflect quality, regardless of density. How is that different under territoriality?
- Line 121: This is a very strong assumption of an absolute priority effect. Is it supported by the empirical data? Is there 0 overlap between MCPs in any given year?
- Line 156-168: I found this section confusing due to a mixture of exponential Habitat Selection Function and logistic Habitat Selection Probability Function terminologies. An exponential HSF is an exponential function, $w(x) = \exp[BH(x)]$, with values proportional to the relative probability of using location x given its habitat value H . All but one of the parameters in the vector B are estimable by fitting a Binomial GLM to binary used/available data. The intercept is not, hence the relative rather than absolute values. In this work, the authors employed a logistic HSPF (Eq. 2), $w(x) = \exp[BH(x)] / (1 + \exp[BH(x)])$, which (given all assumptions are met) yields the actual probability of using x . In this case, All the parameters in the vector B (including the intercept) are estimable by fitting a Binomial GLM to binary used/unused data. Because of the rather unique nature of the data at hand, the authors are able to designate a truly unused domain and fit a HSPF (a logistic one in this case, but it could just as well get any form that integrates to one). If the unused domain could not be safely delineated (because, for example, of uncertainty regarding where the animals were when they were not observed), available points need to be sampled from the available distribution (containing both the used and theoretically

unused but available domains), and a Binomial GLM would result in a fitted exponential HSF (with coefficients that should be interpreted as 'log relative risk' rather than 'log odds'). Other than preventing confusion, this distinction has ramification because of the direct link between the exponential HSF and an inhomogeneous Poisson point process, a link that breaks down for the logistic HSPF. In short, I believe the current contribution focuses on logistic HSPF, and this should be made clear here and elsewhere.

- Line 228-230: This is unclear – it sounds like you already have a full census – why did you need this spatial interpolation procedure?
- Line 244-246: Is there any reason to believe that harvest data reflect deer (or at least buck) abundance rather than hunter abundance (effort)?
- Line 252-253: Why use such a non-parametric approach? Why not choose the best predictor of the two so that the covariate has easily interpretable units?
- Line 260-263: It is entirely unclear why this spatial smoothing was needed and, if it is, why is $\frac{1}{4}$ of a HR the appropriate smoothing scale?
- Line 350: I find it peculiar that the effect of the buck-kill index is negative (Table 2), don't you?
- Line 352: Table 2 reports a negative linear and quadratic effects for slope (so an accelerating avoidance), yet here and in Fig 3 the effect is reported as concave up (so a positive linear effect); which one is it?
- Line 358-360: Perhaps report the AUC for an alternative model that does not contain density-dependent effects (but is otherwise appropriately formulated)?
- Line 361-366: I think whether an effect increases or decreases with density is measured with regards to its intercept (density = 0 or density = mean density?): in the interaction has the same sign as the intercept, the effect increases, but if the interaction has an opposite sign, the effect initially decreases and eventually flips. Interpretation becomes especially tricky when it comes to quadratic effects – the results in Table 2 indicate that the response to elevation for example becomes 'shallower' (flatter parabola) with density, but this is not so clear for 'slope'.

Sincerely,
Tal Avgar.

Decision letter (RSOS-190282.R0)

19-Jun-2019

Dear Dr O'Neil,

The editors assigned to your paper ("Territorial landscapes: incorporating density-dependence into wolf resource selection study designs") have now received comments from reviewers. We would like you to revise your paper in accordance with the referee and Associate Editor suggestions which can be found below (not including confidential reports to the Editor). Please note this decision does not guarantee eventual acceptance.

Please submit a copy of your revised paper before 12-Jul-2019. Please note that the revision deadline will expire at 00.00am on this date. If we do not hear from you within this time then it will be assumed that the paper has been withdrawn. In exceptional circumstances, extensions may be possible if agreed with the Editorial Office in advance. We do not allow multiple rounds of revision so we urge you to make every effort to fully address all of the comments at this stage. If deemed necessary by the Editors, your manuscript will be sent back to one or more of the

original reviewers for assessment. If the original reviewers are not available, we may invite new reviewers.

- Data accessibility

If you wish to submit your supporting data or code to Dryad (<http://datadryad.org/>), or modify your current submission to dryad, please use the following link:
<http://datadryad.org/submit?journalID=RSOS&manu=RSOS-190282>

- Competing interests

- Authors' contributions

- Acknowledgements

- Funding statement

on behalf of Kevin Padian (Subject Editor)
openscience@royalsociety.org

Comments to Author:

Reviewers' Comments to Author:

Reviewer: 1

Comments to the Author(s)

Review of RSOS-190282

O'Neil et al. Territorial landscapes: incorporating density-dependence into wolf resource selection study designs

This manuscript reviews the complexity that density dependence may introduce into both the understanding of habitat selection and attempts to quantify it. The authors then take a use/available approach and use a long-term telemetry data set for wolves in Michigan to explore the nature of density dependence on the drivers of habitat selection over a roughly 20 year period.

This paper is an interesting treatment of an issue that has important theoretical, practical, and policy dimensions – both in terms of what it adds to the ecologist's toolbox (intellectually and practically) and for what it helps us to understand about the expansion of wolves in the Great Lakes region. I think this is an important and worthy piece of research. It is novel and creative and complete in terms of describing the problem and the analytical approach and the reporting of results.

The manuscript is a bit schizophrenic in that it tries to tell two stories concurrently, one about theoretical issue of identifying and density dependence in habitat selection (either generally or in the case of territorial animals) and another about the pattern of wolf expansion in Michigan. I think it can do both because they are complimentary issues (as the authors acknowledge). The ability to tell two stories doesn't require a strict rule about how much content should be allocated to each but the manuscript feels the weakest where one seems story seems missing or not treated as completely as it could be.

Along those lines, conspicuous by its absence is the recent and earlier work done on density dependence in Wisconsin (full disclosure, I was a collaborator on that work) – after all the scientists in Wisconsin were studying the western half of the same population during the same time period. Van Deelen (2009) and Mladenof et al (2009) both speculated about the pattern of settlement and growth stemming from a phenomenon of wolves saturating high-quality habitat first. Indeed Mladenof's series of papers re-doing the habitat model of the Great Lakes looks like an illustration of the pattern that the authors analyze so compellingly here. Similarly, the individual-based model that Stenglein et al (Ecological Modeling) published duplicated the growth pattern seen in the Midwest by making the virtual wolves behave in a habitat quality based pre-emptive pattern. Other papers by Stenglein et al demonstrate density dependence in components of growth over time, that also varied spatially.

Specific comments:

Abstract: The abstract is generally very good. Length is about right given content and the nut of the analysis is well-handled. My quibble is over the concluding sentence. It concludes with a vague statement of territoriality pitched at analysts. I would add a sentence about what it tells us about the biology of wolf recovery in Michigan.

Introduction: The introduction was similarly well done. The problem of failing to deal with the dynamic nature of habitat selection in analyzing for habitat selection is laid out in a compelling way. It could use a paragraph addressing the expansion of Michigan wolves and why habitat selection could be important in that context.

Ln 74: What's "habitat matching"? Is this another term for selection?

Methods: To be honest, the statistical modelling is a bit opaque to me but what I could piece together suggested a thoughtful, if sophisticated approach that tried to deal honestly and realistically with the data and analytical techniques that could be brought to bear. Appendix 2 was very useful in helping to understand the datasets and the treatments that those datasets received to become drivers of use and availability.

Ln 276,277: The sentence structure seems to contrast "territory" with "pack". Aren't these the same thing? Please clarify.

Ln 286,287: Why the imbalance here? Five random locations for use but 25 for available? With small samples could differences between use and availability stem from small sample phenomena?

Results: Well done. Model fit was remarkable!

Discussion: The discussion was also well done although I would recommend that the authors weave a bit more about what they learned about wolf recovery in Michigan as they discuss the theoretical points.

Ln: 444-445 Isn't site dependence a version of density dependence. Perhaps a definition to make things more explicit?

Table 1: Good.

Table 2: I presume the use of bolded text is acceptable for the journal... Otherwise good.

Figure 1: Good.
 Figure 2: Good.
 Figure 3: Good.
 Figure 4: Good.
 Figure 5 Good.
 Figure 6 Good.

Reviewer: 2

Comments to the Author(s)

The MS titled “Territorial landscapes: incorporating density-dependence into wolf resource selection study designs” described an in-depth investigation of a habitat-selection probability function in the presence of territoriality. Overall I found the MS well written and providing the most thorough treatment of density-dependent habitat-selection analysis I know of; it will make a valuable contribution to the scientific literature.

That said, I do have two major(ish) methodological concerns. First, home-range (HR) delineations were done using at least three different approaches: an annual KDE (with variable number of locations as long as >30), 3-year or ‘long-term’ KDE (with variable number of locations as long as >30), or an MCP. Whereas none of these is ‘right’ or ‘wrong’, they may result in very different estimates and have different level of sensitivity to the number of locations and the period over which they were taken. I would be reassuring if the authors could provide a plot comparing HR size as function of these different methods, and a plot showing the relative proportion of each methods used as function of population density. My second, somewhat related, concern is the choice of five random points from the used domain and 25 random locations from the unused domain, regardless of the size of these domains. This means that the size of each home range is completely ignored in the analysis, which may be misleading as HR size might also shift with pack-size, number of neighboring packs, and time (due to familiarity and seniority effects). This also means that the intercept of the HSPF (Eq. 2) is biased. In theory, this intercept would be unbiased only under sampling design in which the numbers of used and unused points are proportional to the area of the respective used and unused domains. In practice, the main effect of population density may, to some degree, correct this bias, but I find it make interpretation extremely difficult. From my perspective, it would make more sense to sample point uniformly from the respective domains and not include a main effect of density. This would also mean that integrating the resulting HSPF over the study area for any specific year should yield (approximately) the total number of wolves occupying the study area in that year, which should provide a nice validation of the model.

Itemized comments:

- Lines 72-75: I think this could use some clarification. Under simple IFD with discrete habitat patches, occupancy is expected to be density dependent (poorer patches get occupied as density increases and they become relatively profitable), but among occupied patches, relative density should reflect quality, regardless of density. How is that different under territoriality?
- Line 121: This is a very strong assumption of an absolute priority effect. Is it supported by the empirical data? Is there 0 overlap between MCPs in any given year?
- Line 156-168: I found this section confusing due to a mixture of exponential Habitat Selection Function and logistic Habitat Selection Probability Function terminologies. An exponential HSF is an exponential function, $w(x) = \exp[BH(x)]$, with values proportional to the relative probability of using location x given its habitat value H . All but one of the parameters in the vector B are estimable by fitting a Binomial GLM to binary used/available data. The intercept is not, hence the relative rather than absolute values. In this work, the authors employed a logistic HSPF (Eq. 2), $w(x) = \exp[BH(x)] / (1 + \exp[BH(x)])$, which (given all assumptions are met) yields the actual probability of using x . In this case, All the parameters in the vector B (including the

intercept) are estimable by fitting a Binomial GLM to binary used/unused data. Because of the rather unique nature of the data at hand, the authors are able to designate a truly unused domain and fit a HSPF (a logistic one in this case, but it could just as well get any form that integrates to one). If the unused domain could not be safely delineated (because, for example, of uncertainty regarding where the animals were when they were not observed), available points need to be sampled from the available distribution (containing both the used and theoretically unused but available domains), and a Binomial GLM would result in a fitted exponential HSF (with coefficients that should be interpreted as 'log relative risk' rather than 'log odds'). Other than preventing confusion, this distinction has ramification because of the direct link between the exponential HSF and an inhomogeneous Poisson point process, a link that breaks down for the logistic HSPF. In short, I believe the current contribution focuses on logistic HSPF, and this should be made clear here and elsewhere.

- Line 228-230: This is unclear – it sounds like you already have a full census – why did you need this spatial interpolation procedure?
- Line 244-246: Is there any reason to believe that harvest data reflect deer (or at least buck) abundance rather than hunter abundance (effort)?
- Line 252-253: Why use such a non-parametric approach? Why not choose the best predictor of the two so that the covariate has easily interpretable units?
- Line 260-263: It is entirely unclear why this spatial smoothing was needed and, if it is, why is $\frac{1}{4}$ of a HR the appropriate smoothing scale?
- Line 350: I find it peculiar that the effect of the buck-kill index is negative (Table 2), don't you?
- Line 352: Table 2 reports a negative linear and quadratic effects for slope (so an accelerating avoidance), yet here and in Fig 3 the effect is reported as concave up (so a positive linear effect); which one is it?
- Line 358-360: Perhaps report the AUC for an alternative model that does not contain density-dependent effects (but is otherwise appropriately formulated)?
- Line 361-366: I think whether an effect increases or decreases with density is measured with regards to its intercept (density = 0 or density = mean density?): in the interaction has the same sign as the intercept, the effect increases, but if the interaction has an opposite sign, the effect initially decreases and eventually flips. Interpretation becomes especially tricky when it comes to quadratic effects – the results in Table 2 indicate that the response to elevation for example becomes 'shallower' (flatter parabola) with density, but this is not so clear for 'slope'.

Sincerely,
Tal Avgar.

Author's Response to Decision Letter for (RSOS-190282.R0)

See Appendix A.

RSOS-190282.R1 (Revision)

Review form: Reviewer 1

Is the manuscript scientifically sound in its present form?

Yes

Are the interpretations and conclusions justified by the results?

Yes

Is the language acceptable?

Yes

Do you have any ethical concerns with this paper?

No

Have you any concerns about statistical analyses in this paper?

No

Recommendation?

Accept as is

Comments to the Author(s)

Nice job. Important work. I have no further comments other than there appears to be a typo in line 262 (page 13).

Review form: Reviewer 2 (Tal Avgar)

Is the manuscript scientifically sound in its present form?

Yes

Are the interpretations and conclusions justified by the results?

Yes

Is the language acceptable?

Yes

Do you have any ethical concerns with this paper?

No

Have you any concerns about statistical analyses in this paper?

No

Recommendation?

Accept with minor revision (please list in comments)

Comments to the Author(s)

The authors have revised the MS thoroughly and thoughtfully, and have done an impressive job at addressing my previous concerns. I believe this MS is now (almost) ready for publication and that it would make a valuable contribution to the literature; I have but a few additional comments and suggestions:

- Lines 146-204: I believe it will increase the MS's readability if the bulk of this would move either to the Introduction or to a separate section between the Introduction and the Methods. I will also help if a more explicit justification is provided for discussing this in the context of RSF but then demonstrating it using an RSPF.

- Lines 183-189: A more careful wording is warranted here. A logistic regression (AKA, GLM with a Binomial link function) with used/available data is used to estimate the parameters of an exponential habitat-selection function at location x with n habitat covariates: $\exp[b_0 + b_1 \cdot h_1(x) + \dots + b_n \cdot h_n(x)]$, where b_0 is a 'nuisance intercept'. This procedure is exponentially equivalent to estimating the relative intensity of an inhomogeneous Poisson spatial point pattern.
- Lines 254-256: perhaps demonstrate this comparison in an appendix figure?
- Line 336 and later use of the term 'available': It seem that analysis is predicated on the assumption that '0' represent true absences, and should thus be refer to as 'unused' rather than 'available'
- Lines 368-370: Based on this, and eq. 2, I understand that the shape fitted RSPF is logistic. It might help to be explicit about this and about the fact that this shape is qualitatively different than the exponential RSF shape discussed earlier; it is important not to confuse the use of logistic regression to estimate the parameters of an exponential RSF, with the fitting a logistic RS(P)F.
- Lines 447-451: Note that the 'effect' of a habitat covariate is 'strong' or 'weak' independent of whether it is positive or negative. For example, according to the reported results, the (negative) effect of stream density increased (became stronger – more negative) with wolf density, whereas the effect of slope (negative) decreased (became weaker – less negative) with wolf density. These interpretations depend on whether 'wolf density' was centered or not; if the former, the main effect of the interacting habitat covariate (e.g., stream density or slope) is estimated assuming mean wolf density, whereas the latter implies, the main effect of the interacting habitat covariate (is estimated assuming wolf density = 0.
- Lines 494-502: perhaps add a sentence or two about the potential demographic dynamics that might emerge from this
- Lines 559-560: I have to disagree – surprisingly little attention has been given to density-dependent habitat selection patterns, both on the empirical and theoretical sides.
- Table 1: I would be curious to know what is the biological interpretation of the main effect of wolf density (0.724)
- Fig 4: it might be just my 3D-challenged perception, but I find these very difficult to interpret

Sincerely,
Tal Avgar.

Decision letter (RSOS-190282.R1)

08-Oct-2019

Dear Dr O'Neil:

On behalf of the Editors, I am pleased to inform you that your Manuscript RSOS-190282.R1 entitled "Territorial landscapes: incorporating density-dependence into wolf resource selection study designs" has been accepted for publication in Royal Society Open Science subject to minor revision in accordance with the referee suggestions. Please find the referees' comments at the end of this email.

The reviewers and Subject Editor have recommended publication, but also suggest some minor revisions to your manuscript. Therefore, I invite you to respond to the comments and revise your manuscript.

- Ethics statement

If your study uses humans or animals please include details of the ethical approval received, including the name of the committee that granted approval. For human studies please also detail

whether informed consent was obtained. For field studies on animals please include details of all permissions, licences and/or approvals granted to carry out the fieldwork.

- Data accessibility

If you wish to submit your supporting data or code to Dryad (<http://datadryad.org/>), or modify your current submission to dryad, please use the following link:
<http://datadryad.org/submit?journalID=RSOS&manu=RSOS-190282.R1>

- Competing interests

- Authors' contributions

- Acknowledgements

- Funding statement

Because the schedule for publication is very tight, it is a condition of publication that you submit the revised version of your manuscript before 17-Oct-2019. Please note that the revision deadline

will expire at 00.00am on this date. If you do not think you will be able to meet this date please let me know immediately.

Kind regards,
Lianne Parkhouse
Royal Society Open Science
openscience@royalsociety.org

on behalf of the Associate Editor, and Professor Kevin Padian (Subject Editor)
openscience@royalsociety.org

Associate Editor Comments to Author:

Thanks for making such strong efforts to complete the revisions of your paper - only a few outstanding comments appear to remain from the reviewers before acceptance is possible. Please make sure these tweaks are incorporated into your final revision. Good luck!

Reviewer comments to Author:

Reviewer: 1

Comments to the Author(s)

Nice job. Important work. I have no further comments other than there appears to be a typo in line 262 (page 13).

Reviewer: 2

Comments to the Author(s)

The authors have revised the MS thoroughly and thoughtfully, and have done an impressive job at addressing my previous concerns. I believe this MS is now (almost) ready for publication and that it would make a valuable contribution to the literature; I have but a few additional comments and suggestions:

- Lines 146-204: I believe it will increase the MS's readability if the bulk of this would move either to the Introduction or to a separate section between the Introduction and the Methods. I will also help if a more explicit justification is provided for discussing this in the context of RSF but then demonstrating it using an RSPF.
- Lines 183-189: A more careful wording is warranted here. A logistic regression (AKA, GLM with a Binomial link function) with used/available data is used to estimate the parameters of an exponential habitat-selection function at location x with n habitat covariates: $\exp[b_0 + b_1 h_1(x) + \dots + b_n h_n(x)]$, where b_0 is a 'nuisance intercept'. This procedure is exponentially equivalent to estimating the relative intensity of an inhomogeneous Poisson spatial point pattern.
- Lines 254-256: perhaps demonstrate this comparison in an appendix figure?
- Line 336 and later use of the term 'available': It seem that analysis is predicated on the assumption that '0' represent true absences, and should thus be refer to as 'unused' rather than 'available'
- Lines 368-370: Based on this, and eq. 2, I understand that the shape fitted RSPF is logistic. It might help to be explicit about this and about the fact that this shape is qualitatively different than the exponential RSF shape discussed earlier; it is important not to confuse the use of logistic regression to estimate the parameters of an exponential RSF, with the fitting a logistic RS(P)F.
- Lines 447-451: Note that the 'effect' of a habitat covariate is 'strong' or 'weak' independent of whether it is positive or negative. For example, according to the reported results, the (negative) effect of stream density increased (became stronger - more negative) with wolf density, whereas the effect of slope (negative) decreased (became weaker - less negative) with wolf density. These interpretations depend on whether 'wolf density' was centered or not; if the former, the main effect of the interacting habitat covariate (e.g., stream density or slope) is estimated assuming mean wolf density, whereas the latter implies, the main effect of the interacting habitat covariate (is estimated assuming wolf density = 0).
- Lines 494-502: perhaps add a sentence or two about the potential demographic dynamics that might emerge from this
- Lines 559-560: I have to disagree - surprisingly little attention has been given to density-dependent habitat selection patterns, both on the empirical and theoretical sides.

- Table 1: I would be curious to know what is the biological interpretation of the main effect of wolf density (0.724)
- Fig 4: it might be just my 3D-challenged perception, but I find these very difficult to interpret

Sincerely,
Tal Avgar.

Author's Response to Decision Letter for (RSOS-190282.R1)

See Appendix B.

Decision letter (RSOS-190282.R2)

16-Oct-2019

Dear Dr O'Neil,

I am pleased to inform you that your manuscript entitled "Territorial landscapes: incorporating density-dependence into wolf resource selection study designs" is now accepted for publication in Royal Society Open Science.

Kind regards,

on behalf of the Associate Editor, and Professor Kevin Padian (Subject Editor)
openscience@royalsociety.org

Appendix A

21-Sep-2019 (Author response & summary)

Dear Editors of Royal Society Open Science,

We have completed our revision of RSOS-190282, *Territorial landscapes: incorporating density-dependence into wolf resource selection study designs*. We felt that the reviewers had excellent insights and suggestions, and in response, we have made major revisions to our analysis and presentation of the manuscript. We believe that we have accommodated nearly every suggestion; if we could not accommodate a suggestion, we provide justification as to our reasoning. I have provided an outline of all the major analytical revisions that were made below. We have restructured and revised our text accordingly. We also incorporated more biological context for density dependent population dynamics in Great Lakes wolves into the Introduction and Discussion, with appropriate references. We hope you will find the manuscript to be much improved, and now more suitable for publication in Royal Society Open Science. Thank you for your time and consideration.

1) Random sampling design and used vs. available/unoccupied locations

Both reviewers expressed confusion as to why we had initially sampled evenly with a ratio of 5:25 in used vs. available/unoccupied locations. While there may be valid reason for this, we agreed with Dr. Avgar (Reviewer 2) that it was more logical to sample more proportionally to area occupied vs. unoccupied from the wolf pack's perspective. This required a thorough revision of the analysis, starting from the beginning stages of the analysis. As a result, much of the results and discussion have undergone at least some revision because the output from the model has changed (although the main findings are generally consistent with the previous version).

2) Incorporating territory overlap and uncertainty in occupied area

First, the 2nd reviewer questioned the assumption of complete competitive exclusion. This was a valid criticism, and we dealt with it by allowing a small proportion of "available" points to fall within occupied area from each pack's perspective. In fact, we now incorporate observed changes in the overall proportion of territory overlap over time and allow this proportion to inform how many available random points fall within potentially occupied area. This may create some additional challenges in interpretation, but we felt that it was more representative of the true population dynamics, and an important component of density dependence that our analysis was previously lacking. We have added discussion and explanation accordingly.

Second, because we could not observe territory boundaries with high precision, we developed an inverse probability of occupancy surface that accounts for uncertainty in pack occupancy from one year to the next based on the variation in territory size within pack (for telemetry territory home ranges) and among packs (for MCPs of ground tracking territory home ranges). We used this surface to govern where unoccupied/available locations were randomly drawn from, using a spatially balanced random point design.

We have combined the original Figures 5 & 6 into a new figure (Figure 2) that also includes changes in territory overlap with density.

3) Removing buck kill index from the analysis

Reviewer 2 pointed out some concerns with using buck kill as an index for deer density. Ultimately, we decided it was best not to include this predictor, as it occurs at a much coarser resolution than other predictors and comprises temporal variation that may not be associated with fluctuations in actual deer abundance, thereby confounding interpretation. In addition, it cannot be separated from hunter effort.

4) Removing consideration of quadratic effects for topography variables

We removed quadratic terms for elevation and slope because these seemed to be excessive and confusing parameters given that our model already includes random pack coefficients (e.g., likely overfitting at the pack level when including pack-specific coefficient, density interaction, and quadratic term).

Please see our detailed responses to each reviewer comment below.

Thank you for considering our revision.

Sincerely,

Shawn O'Neil

19-Jun-2019

Dear Dr O'Neil,

The editors assigned to your paper ("Territorial landscapes: incorporating density-dependence into wolf resource selection study designs") have now received comments from reviewers. We would like you to revise your paper in accordance with the referee and Associate Editor suggestions which can be found below (not including confidential reports to the Editor). Please note this decision does not guarantee eventual acceptance.

Please submit a copy of your revised paper before 12-Jul-2019. Please note that the revision deadline will expire at 00.00am on this date. If we do not hear from you within this time then it will be assumed that the paper has been withdrawn. In exceptional circumstances, extensions may be possible if agreed with the Editorial Office in advance. We do not allow multiple rounds of revision so we urge you to make every effort to fully address all of the comments at this stage. If deemed necessary by the Editors, your manuscript will be sent back to one or more of the original reviewers for assessment. If the original reviewers are not available, we may invite new reviewers.

- Data accessibility

<http://datadryad.org/submit?journalID=RSOS&manu=RSOS-190282>

- Competing interests

- Authors' contributions

- Acknowledgements

- Funding statement

Kind regards,

Alice Power

Editorial Coordinator

on behalf of Kevin Padian (Subject Editor)

Comments to Author:

Reviewers' Comments to Author:

Reviewer: 1

Comments to the Author(s)

Review of RSOS-190282

O'Neil et al. Territorial landscapes: incorporating density-dependence into wolf resource selection study designs

This manuscript reviews the complexity that density dependence may introduce into both the understanding of habitat selection and attempts to quantify it. The authors then take a use/available approach and use a long-term telemetry data set for wolves in Michigan to explore the nature of density dependence on the drivers of habitat selection over a roughly 20 year period.

This paper is an interesting treatment of an issue that has important theoretical, practical, and policy dimensions – both in terms of what it adds to the ecologist's toolbox (intellectually and practically) and for what it helps us to understand about the expansion of wolves in the Great Lakes region. I think this is an important and worthy piece of research. It is novel and creative and complete in terms of describing the problem and the analytical approach and the reporting of results.

Agreed, glad to hear positive feedback.

The manuscript is a bit schizophrenic in that it tries to tell two stories concurrently, one about theoretical issue of identifying and density dependence in habitat selection (either generally or in the case of territorial animals) and another about the pattern of wolf expansion in Michigan. I think it can do both because they are complimentary issues (as the authors acknowledge). The ability to tell two stories doesn't require a strict rule about how much content should be allocated to each but the manuscript feels the weakest where one seems story seems missing or not treated as completely as it could be.

While this is somewhat difficult to interpret, we believe the following statements are helpful in elaborating, so we respond and revise primarily based on the more specific comments below.

Along those lines, conspicuous by its absence is the recent and earlier work done on density dependence in Wisconsin (full disclosure, I was a collaborator on that work) – after all the scientists in Wisconsin were studying the western half of the same population during the same time period. Van Deelen (2009) and Mladenoff et al (2009) both speculated about the pattern of settlement and growth stemming from a phenomenon of wolves saturating high-quality habitat first. Indeed Mladenoff's series of papers re-doing the habitat model of the Great Lakes looks like an illustration of the pattern that the authors analyze so compellingly here. Similarly, the individual-based model that Stenglein et al (Ecological Modeling) published duplicated the growth pattern seen in the Midwest by making the virtual wolves behave in a habitat quality based pre-emptive pattern. Other papers by Stenglein et al demonstrate density dependence in components of growth over time, that also varied spatially.

We agree – the works by Van Deelen, Mladenoff and Stenglein all suggest a pre-emptive habitat selection type of strategy relevant to expansion of wolves in the Great Lakes region and should have been referenced. Omission of these papers was an unintentional mistake and was likely due to our heavy focus on habitat modeling. We have corrected this unintentional omission by adding some discussion of

our work in the context of other relevant findings with respect to Great Lakes wolves (Introduction, lines 104–111, Discussion, lines 489-492).

Specific comments:

Abstract: The abstract is generally very good. Length is about right given content and the nut of the analysis is well-handled. My quibble is over the concluding sentence. It concludes with a vague statement of territoriality pitched at analysts. I would add a sentence about what it tells us about the biology of wolf recovery in Michigan.

Good idea, we have incorporated this suggestion into the end of the abstract.

Introduction: The introduction was similarly well done. The problem of failing to deal with the dynamic nature of habitat selection in analyzing for habitat selection is laid out in a compelling way. It could use a paragraph addressing the expansion of Michigan wolves and why habitat selection could be important in that context.

Thank you for this suggestion, we've incorporated additional description of wolf expansion to Michigan in the context of Great Lakes wolf recovery, with reference to the works that we had previously mistakenly omitted. That context is indeed important, and its inclusion will hopefully strengthen the manuscript and should make it more accessible to the larger audience following wolf expansion in the U.S. and Europe.

Ln 74: What's "habitat matching"? Is this another term for selection?

Because multiple reviewers found this confusing, we omit usage of the term "habitat matching" in the revision. Habitat matching refers to density-dependent habitat selection, where lower quality habitats are selected as a function of density in higher quality habitats (the nature of this depends on ideal-free vs. ideal-dominant distributed populations). However, it is unnecessary and confusing here and would require substantial theoretical explanation that we feel goes beyond the scope of this Introduction.

Methods: To be honest, the statistical modelling is a bit opaque to me but what I could piece together suggested a thoughtful, if sophisticated approach that tried to deal honestly and realistically with the data and analytical techniques that could be brought to bear. Appendix 2 was very useful in helping to understand the datasets and the treatments that those datasets received to become drivers of use and availability.

We appreciate this feedback. Because there are no specific suggestions here, we have not made any corresponding changes in response to these comments. However, we have made significant revisions to the methods in response to other reviewers, so bear in mind some textual changes were necessary. We hope these changes go towards making the modelling less opaque.

Ln 276,277: The sentence structure seems to contrast "territory" with "pack". Aren't these the same thing? Please clarify.

Apologies for the confusing language, this contrast was unintentional. We've corrected this sentence to avoid further confusion. In the new revised draft, this section now starts on Line 336. Thanks for catching it!

Ln 286,287: Why the imbalance here? Five random locations for use but 25 for available? With small samples could differences between use and availability stem from small sample phenomena?

The imbalance was based on attempts to follow guidelines within habitat selection modeling literature suggesting that these types of analyses should weight available locations more heavily than used locations; however, because we are assuming availability generally represents absence, this is not necessary, as pointed out by another reviewer, and we've abandoned that sampling imbalance in

response to multiple reviewer suggestions. The section 'Sampling Design and Resource Selection Probability Functions' has been modified throughout to reflect these changes.

We would also like to clarify that small sample should not be an issue here, as inference is drawn at the population level, where the number of locations drawn from used vs. available domains is >1000 annually. The sampling scheme now draws used vs. available samples proportional to overall area occupied, such that the intercept-only model approximates the overall proportion of area occupied. This was done based on recommendations from another reviewer.

Results: Well done. Model fit was remarkable!

Thanks! One reason the model fit this well was because we were able to estimate pack-specific coefficients for the different habitat types. This is a major advantage of this data set and key strength of this overall analysis. This also allows for exploration of pack-level fitness related to selection patterns – we would like to explore this further at some point, perhaps if there is a dataset with some quality information on lifetime reproductive success (or something similar).

Discussion: The discussion was also well done although I would recommend that the authors weave a bit more about what they learned about wolf recovery in Michigan as they discuss the theoretical points.

Thank you for the suggestion, we have added further discussion of MI wolf recovery in the 2nd and 3rd paragraphs of the Discussion.

Ln: 444-445 Isn't site dependence a version of density dependence. Perhaps a definition to make things more explicit?

Good point – we've attempted to define and clarify (Lines 536–542)

Table 1: Good.

Table 2: I presume the use of bolded text is acceptable for the journal... Otherwise good.

Figure 1: Good.

Figure 2: Good.

Figure 3: Good.

Figure 4: Good.

Figure 5 Good.

Figure 6 Good.

Reviewer: 2

Comments to the Author(s)

The MS titled "Territorial landscapes: incorporating density-dependence into wolf resource selection study designs" described an in-depth investigation of a habitat-selection probability function in the presence of territoriality. Overall I found the MS well written and providing the most thorough treatment of density-dependent habitat-selection analysis I know of; it will make a valuable contribution to the scientific literature.

We're glad to hear positive feedback and appreciate the support.

That said, I do have two major(ish) methodological concerns. First, home-range (HR) delineations were done using at least three different approaches: an annual KDE (with variable number of locations as long as >30), 3-year or 'long-term' KDE (with variable number of locations as long as >30), or an MCP. Whereas none of these is 'right' or 'wrong', they may result in very different estimates and have different level of sensitivity to the number of locations and the period over which they were taken. I would be reassuring if the authors could provide a plot comparing HR size as function of these different methods,

We've generated this plot and added it to the Supplementary Appendices (Appendix S2) with some explanation. There is substantial variation in home range (territory) size, regardless of the method used. There is some mild evidence that the long-term KDE may result in a slightly larger territory size than the other two methods. However, the predicted difference from a GLM incorporating time and pack size (Area = year + pack size + method) is ~ 20 km², which is reasonable considering the overall average territory size (~283 km²) and amount of natural variation in territory size based on the annual KDE (e.g., best case scenario).

Given that in this revision we are now relaxing the assumption of complete exclusion (see responses to other comments, and edits made to the methods), and also quantifying uncertainty with respect to the available, unoccupied domain (see new section 'uncertainty in territory occupancy'), we feel that any bias associated with these differences is likely minimal.

and a plot showing the relative proportion of each methods used as function of population density.

This comment is unclear to us. Unfortunately, we don't quite follow how this plot would be constructed; the methods used were not a function of population density. The methods employed are likely a function of project funding and logistical constraints, which influenced the number of packs monitored with collared wolves on an annual basis, as well as the frequency of aerial flights. Funding was variable, related to national Endangered Species Act decisions and funding at the federal level as well as changes in political administration that tend to influence Pittman-Robertson funds.

My second, somewhat related, concern is the choice of five random points from the used domain and 25 random locations from the unused domain, regardless of the size of these domains. This means that the size of each home range is completely ignored in the analysis, which may be misleading as HR size might also shift with pack-size, number of neighboring packs, and time (due to familiarity and seniority effects). This also means that the intercept of the HSPF (Eq. 2) is biased. In theory, this intercept would be unbiased only under sampling design in which the numbers of used and unused points are proportional to the area of the respective used and unused domains. In practice, the main effect of population density may, to some degree, correct this bias, but I find it make interpretation extremely difficult. From my perspective, it would make more sense to sample point uniformly from the respective domains and not include a main effect of density.

Our initial motivation for equal sampling between used & available domains for each pack was stemming from RSF modelling literature, where the availability distribution should be weighted more heavily than the used distribution (in fact, as heavily weighted as possible, see for example Northrup et al. 2013, Ecology). In addition, we found it appealing to have each pack equally weighted in terms of probability of selection, for coefficients to be interpreted as a 2nd order scale of selection. However, as you point out, we are assuming that the available distribution represents true unoccupied range, so the sampling recommendations for HSF (as opposed to HSPF), and perhaps even the desire for packs to be equally weighted, do not necessarily apply.

Admittedly, distinguishing our approach from a HSF analysis (as you have commented on below) is nuanced, given that one of our objectives is to bring this issue to the attention of those that may not have data on true absence (but are dealing with territorial individuals), because that is perhaps the most common circumstance for managers and applied ecologists studying populations of large carnivores. We have attempted to clarify the distinction throughout.

*All of that said, and back to the point, we were intrigued by your suggestion to sample uniformly within used & available distributions for each pack, thereby taking into consideration the size of the home range. We have set up a new analysis using this sampling design. We describe in detail in the sections '**Sampling Design and Resource Selection Probability Functions.**' We hope we have interpreted your suggestions correctly, and that this design is more consistent with your perspective.*

Reference: Northrup, J. M., Hooten, M. B., Anderson Jr, C. R., & Wittemyer, G. (2013). Practical guidance on characterizing availability in resource selection functions under a use–availability design. Ecology, 94(7), 1456-1463.

This would also mean that integrating the resulting HSPF over the study area for any specific year should yield (approximately) the total number of wolves occupying the study area in that year, which should provide a nice validation of the model.

Although we think we understand this latter idea in concept, we are not sure how the result would yield an estimate of the number of wolves, given that the response variable represents, effectively, the probability of occupancy, i.e. that any randomly selected point location in the sample belongs to the occupied vs. unoccupied domain. It does not take into consideration pack sizes (other than through interactions of density with the habitat covariates, that are indirectly related to pack size).

Itemized comments:

- Lines 72-75: I think this could use some clarification. Under simple IFD with discrete habitat patches, occupancy is expected to be density dependent (poorer patches get occupied as density increases and they become relatively profitable), but among occupied patches, relative density should reflect quality, regardless of density. How is that different under territoriality?

Thank you for the suggestion, we agree, and have added additional clarification to the beginning of the third paragraph, lines 75–83. Specifically, we now point out that density may not reflect quality under IDD or IPD distributions.

- Line 121: This is a very strong assumption of an absolute priority effect. Is it supported by the empirical data? Is there 0 overlap between MCPs in any given year?

*This is a valid point. We explored pack territory overlap further and found some interesting results. Overlap was minimal early in the study (< 2% total area overlap), but increased more than we expected later in the study (as high as 48% total area overlap). Clearly this needed some sort of treatment, so we updated our sampling design so that a proportion of “available/unused” points were allowed to fall into the occupied domain. This proportion was based on an estimate of territory area overlap that we repeated on an annual timestep. The new methods describe this in more detail in the section ‘**Sampling Design and Resource Selection Probability Functions.**’ Thank you for bringing it to our attention, as we feel it significantly improves our paper.*

- Line 156-168: I found this section confusing due to a mixture of exponential Habitat Selection Function and logistic Habitat Selection Probability Function terminologies. An exponential HSF is not an exponential function, $w(x) = \exp[BH(x)]$, with values proportional to the relative probability of using location x given its habitat value H . All but one of the parameters in the vector B are estimable by fitting a Binomial GLM to binary used/available data. The intercept is not, hence the relative rather than absolute values. In this work, the authors employed a logistic HSPF (Eq. 2), $w(x) = \exp[BH(x)] / (1 + \exp[BH(x)])$, which (given all assumptions are met) yields the actual probability of using x . In this case, All the parameters in the vector B (including the intercept) are estimable by fitting a Binomial GLM to binary used/unused data. Because of the rather unique nature of the data at hand, the authors are able to designate a truly unused domain and fit a HSPF (a logistic one in this case, but it could just as well get any form that integrates to one). If the unused domain could not be safely delineated (because, for example, of uncertainty regarding where the animals were when they were not observed), available points need to be sampled from the available distribution (containing both the used and theoretically unused but available domains), and a Binomial GLM would result in a fitted exponential HSF (with coefficients that should be interpreted as ‘log relative risk’ rather than ‘log odds’). Other than preventing confusion, this distinction has ramification because of the direct link between the exponential HSF and an

inhomogeneous Poisson point process, a link that breaks down for the logistic HSPF. In short, I believe the current contribution focuses on logistic HSPF, and this should be made clear here and elsewhere.

Thank you for this comment, it is correct that we employ an RPSF/HSPF approach. We did not intend to be confusing, but we still feel that it is important to point out the potential issues and challenges that are brought forth when considering an RSF/HSF analysis for territorial species. For that reason, we have expanded on this section to provide additional clarification and emphasize that while we ultimately employ the HSPF, it is useful to consider the problem of territoriality from a use/availability or RSF perspective. Specifically, see lines 167–204.

- Line 228-230: This is unclear – it sounds like you already have a full census – why did you need this spatial interpolation procedure?

We smoothed the pack-level wolf density estimates to obtain a regional estimate of wolf density. For example, the functional response in habitat selection conditions local estimates of habitat coefficients on their regional availabilities. In this case, we are using density to test for changes in habitat selection conditional on density-dependent changes in regional availability of habitat covariates, so we perform the smoothing to approximate regional, as opposed to local, wolf densities. We've updated the text [lines 262–268] to provide further clarification.

- Line 244-246: Is there any reason to believe that harvest data reflect deer (or at least buck) abundance rather than hunter abundance (effort)?

We had some reason to believe that the buck kill index reflects buck, and likely deer, abundance. This is based on buck density values obtained from camera trapping. However, after much deliberation, we decided to remove the buck harvest index due to issues of spatial resolution (coarse scale, not matching other predictors), hunter effort, and confounding variation over time that might not match true changes in deer density.

- Line 252-253: Why use such a non-parametric approach? Why not choose the best predictor of the two so that the covariate has easily interpretable units?

The PCA approach allowed us to combine predictors and use the information from both; this has worked well in other applications with a larger number of correlated predictors. However, in this case, we agree that with a small number of predictors that offer similar information, it is more desirable to have easily interpretable units. We've made the change in our revision, selecting just the distance to deer wintering complex metric based on the preliminary analysis using Lasso.

- Line 260-263: It is entirely unclear why this spatial smoothing was needed and, if it is, why is $\frac{1}{4}$ of a HR the appropriate smoothing scale?

We've added clarification in the text [lines 319–327]. Spatial smoothing was necessary to evaluate predictors at a scale that was consistent with wolves' selection of territory home ranges. For example, estimation of road density requires a distance over area calculation, where area must be specified. We chose to evaluate predictors using a circular assessment window of $\frac{1}{4}$ of mean wolf home range because we considered this to be a compromise between scales of habitat selection, where within-territory selection patterns are at least partially represented despite performing a landscape-scale analysis.

- Line 350: I find it peculiar that the effect of the buck-kill index is negative (Table 2), don't you?

As explained in our response to your previous comment on buck-kill index, we decided there were too many issues with it to justify its inclusion in the analysis. We believe that winter prey is probably more important, and that other variables such as stream density and forest-open edge density are likely proxies for summer deer densities.

- Line 352: Table 2 reports a negative linear and quadratic effects for slope (so an accelerating avoidance), yet here and in Fig 3 the effect is reported as concave up (so a positive linear effect); which one is it?

We ultimately decided that the quadratic effects for slope and elevation were confusing, excessive parameters, given that we include random slope coefficients for predictors and that we could not consider similar non-linear effects for other predictors. For these reasons, we no longer consider and report quadratic effects.

- Line 358-360: Perhaps report the AUC for an alternative model that does not contain density-dependent effects (but is otherwise appropriately formulated)?

Done, see lines 441–446.

- Line 361-366: I think whether an effect increases or decreases with density is measured with regards to its intercept (density = 0 or density = mean density?): in the interaction has the same sign as the intercept, the effect increases, but if the interaction has an opposite sign, the effect initially decreases and eventually flips. Interpretation becomes especially tricky when it comes to quadratic effects – the results in Table 2 indicate that the response to elevation for example becomes ‘shallower’ (flatter parabola) with density, but this is not so clear for ‘slope’.

We decided to remove consideration of quadratic effects, see response to previous comment.

Sincerely,
Tal Avgar.

Journal Name: Royal Society Open Science

Journal Code: RSOS

Online ISSN: 2054-5703

Journal Admin Email: openscience@royalsociety.org

Journal Editor: Andrew Dunn

Journal Editor Email: openscience@royalsociety.org

MS Reference Number: RSOS-190282

Article Status: SUBMITTED

MS Dryad ID: RSOS-190282

MS Title: Territorial landscapes: incorporating density-dependence into wolf resource selection study designs

MS Authors: O'Neil, Shawn; Beyer, Dean; Bump, Joseph

Contact Author: Shawn O'Neil

Contact Author Email: oneil.shawnt@gmail.com

Contact Author Address 1:

Contact Author Address 2:

Contact Author Address 3:

Contact Author City: Houghton

Contact Author State: Michigan

Contact Author Country: United States

Contact Author ZIP/Postal Code: 49931-1295

Keywords: density-dependence, predator-prey, source-sink population dynamics, species distribution model, habitat selection functional response, carnivores

Abstract: Habitat selection is a process that spans space, time, and individual life histories. Ecological analyses of animal distributions and preferences are most accurate when they account for inherent dynamics of the habitat selection process. Since habitat selection is a function of habitat availability,

strong territoriality can constrain the habitat perceived to be available to individual animals or groups attempting to colonize or establish new territory. When considering a population change over time, broad-scale changes in habitat availability can drive density-dependent variation in habitat selection. We investigated density-dependent habitat selection over a 19-year period of gray wolf (*Canis lupus*) recovery in Michigan, USA using a generalized linear mixed model (GLMM) framework with habitat selection coefficients conditioned on random effects for wolf packs and random year intercepts (e.g. crossed random effects). In addition, we allowed habitat selection coefficients to vary as interactions with increasing wolf density over space and time. Results indicated that the probability of pack occupancy was driven largely by winter prey availability and human impact indices, but that selection coefficients for multiple predictors were density-dependent. Density-dependent habitat selection models had good fit to pack occupancy data, but changes in occupancy at the landscape level tracked changes in used and availability distributions more explicitly across time. Spatiotemporal dynamics and population changes can cause considerable variation in wildlife-habitat relationships. We encourage modelers to adopt flexible approaches to account for potential influences of territoriality when applying traditional habitat selection procedures.

EndDryadContent

Appendix B

Associate Editor Comments to Author:

Thanks for making such strong efforts to complete the revisions of your paper - only a few outstanding comments appear to remain from the reviewers before acceptance is possible. Please make sure these tweaks are incorporated into your final revision. Good luck!

Thank you for this opportunity, we are pleased to submit our final revision. We've accommodated these outstanding comments as indicated in our responses below. We've uploaded a new, clean version of the manuscript, with the final revisions indicated by line numbers associated with our responses. We look forward to hearing from you.

Reviewer comments to Author:

Reviewer: 1

Comments to the Author(s)

Nice job. Important work. I have no further comments other than there appears to be a typo in line 262 (page 13).

Thanks again. We have corrected the typo on (previously) line 262.

Reviewer: 2

Comments to the Author(s)

The authors have revised the MS thoroughly and thoughtfully, and have done an impressive job at addressing my previous concerns. I believe this MS is now (almost) ready for publication and that it would make a valuable contribution to the literature;

We are glad to hear that our efforts have improved the manuscript. We address the additional comments and suggestions below and within the revised manuscript.

I have but a few additional comments and suggestions:

- Lines 146-204: I believe it will increase the MS's readability if the bulk of this would move either to the Introduction or to a separate section between the Introduction and the Methods. I will also help if a more explicit justification is provided for discussing this in the context of RSF but then demonstrating it using an RSPF.

We've addressed this suggestion by moving 'Methodological background and framework' before the methods section (lines 126–148). We defer to the editors regarding whether or not a sub-heading is appropriate in this location, but we do agree that the foundational description is a better fit in this location.

We also provide the requested justification for describing RSFs and contrasting with an RSPF application in the sentences prior to this section (lines 120–124), as well as reiteration and additional justification, lines 142–146.

- Lines 183-189: A more careful wording is warranted here. A logistic regression (AKA, GLM with a Binomial link function) with used/available data is used to estimate the parameters of an

exponential habitat-selection function at location x with n habitat covariates: $\exp[b_0 + b_1 \cdot h_1(x) + \dots + b_n \cdot h_n(x)]$, where b_0 is a 'nuisance intercept'. This procedure is exponentially equivalent to estimating the relative intensity of an inhomogeneous Poisson spatial point pattern.

Thank you for the suggestion, we have expanded this description with more specific text and clarification. Due to reorganization elsewhere, this change is now evident on lines 157–166.

- Lines 254-256: perhaps demonstrate this comparison in an appendix figure?

We have incorporated this figure into the appendix (5), with appropriate reference in the results section

- Line 336 and later use of the term 'available': It seem that analysis is predicated on the assumption that '0' represent true absences, and should thus be refer to as 'unused' rather than 'available'

Agreed, we have made the correction here and throughout.

- Lines 368-370: Based on this, and eq. 2, I understand that the shape fitted RSPF is logistic. It might help to be explicit about this and about the fact that this shape is qualitatively different than the exponential RSF shape discussed earlier; it is important not to confuse the use of logistic regression to estimate the parameters of an exponential RSF, with the fitting a logistic RS(P)F.

Thanks for the suggestion, we have qualified this distinction by contrasting it with the RSF (exponential link) and referencing the earlier section (lines 369–376). We hope this helps to clarify.

- Lines 447-451: Note that the 'effect' of a habitat covariate is 'strong' or 'weak' independent of whether it is positive or negative. For example, according to the reported results, the (negative) effect of stream density increased (became stronger – more negative) with wolf density, whereas the effect of slope (negative) decreased (became weaker – less negative) with wolf density. These interpretations depend on whether 'wolf density' was centered or not; if the former, the main effect of the interacting habitat covariate (e.g., stream density or slope) is estimated assuming mean wolf density, whereas the latter implies, the main effect of the interacting habitat covariate (is estimated assuming wolf density = 0.

We have noted this and corrected the misleading interpretations (see lines 444–470). Because wolf density was centered, the main effects of other predictors are conditioned on wolf density. We have made sure to clarify this in the previous paragraph and in the caption for Table 1.

- Lines 494-502: perhaps add a sentence or two about the potential demographic dynamics that might emerge from this

Thanks for the suggestion, we have elaborated on these implications throughout the paragraph (lines 509–520).

- Lines 559-560: I have to disagree – surprisingly little attention has been given to density-dependent habitat selection patterns, both on the empirical and theoretical sides.

Yes, “wide recognition” is probably an overstatement. We replaced with “some” recognition (line 576–578), which is probably more accurate. We agree that in the context of habitat selection studies, these phenomena have received less research attention than deserved.

- Table 1: I would be curious to know what is the biological interpretation of the main effect of wolf density (0.724)

Following from McLoughlin et al. 2010, section ‘Density dependent habitat selection’, the main effect & interactions with wolf density account for the increasing number of sites that will be occupied at greater population sizes (main effect), as well as the habitat variation in the sites that become occupied as density fluctuates (interactions), because packs/individuals will be searching for sites that yield maximum fitness. We think the main effect of density can be perceived as a nuisance parameter if one is not concerned about the relationship between density and number of occupied sites (which may be of interest to some).

Reference: McLoughlin, P.D., D.W. Morris, et al. (2010) Considering ecological dynamics in resource selection functions. Journal of Animal Ecology 79:4-12.

- Fig 4: it might be just my 3D-challenged perception, but I find these very difficult to interpret

We apologize for any difficulty in perception, but we felt it was important to show the interactions on a continuous gradient of wolf density, hence the inclusion of the 3rd axis for wolf density. This allowed visualization of the gradient of probability selection that was more consistent with model results, as opposed to binning wolf densities and showing effects in 2D.

Sincerely,
Tal Avgar